# Bootstrapped Exploration with Causal Reasoning: A Training Paradigm for Adaptive Forecasting Agent

Qingwen Zeng [* 1]   Dajun Guo [* 2]   Zhaoge Bi [* 1]   Lining Chen [1]   Jushang Qiu [3]   Yitian Yang [1]   Carl Yang [4]
Huaming Chen [1]   Ling Chen [5]

## Abstract

Time series forecasting is critical in domains such as finance, energy, and healthcare, yet real-world datasets often exhibit non-stationarity, noise, missing values, and distribution shifts, posing severe challenges for generalization. In practice, industry solutions typically rely on customized forecasting frameworks that combine imputation, decomposition, and specialized models. However, such frameworks are costly to engineer and maintain. Moreover, we observe that many frameworks suffer from the impacts of distribution shifts, which degrade their respective performance. It motivates a paradigm that transfers reliably across heterogeneous datasets while accumulating reusable strategy knowledge for large-scale, dynamic environments. Although large language model-based agents have recently shown strong reasoning and tool-use capabilities, existing approaches do not consistently adapt forecasting workflows across diverse time series. We identify two primary factors, including limited strategy-level supervision and the inherent complexity of mapping dataset-specific meta-features to effective forecasting strategies. To address these challenges, we propose **BECRA**, a novel agent training paradigm that learns forecasting intelligence through contrast-aware exploration and agent-level causal lesson extraction, without human-annotated supervision. BECRA distills symbolic strategy lessons that support in-context planning on unseen datasets, enabling zero-shot training adaptation.

---

*Equal contribution   [1]The University of Sydney, Sydney, Australia   [2]University of Melbourne, Melbourne, Australia   [3]Australian National University, Canberra, Australia   [4]Emory University, Atlanta, USA   [5]University of Technology Sydney, Sydney, Australia. Correspondence to: Huaming Chen <huaming.chen@sydney.edu.au>.

*Proceedings of the 43$^{rd}$ International Conference on Machine Learning*, Seoul, South Korea. PMLR 306, 2026. Copyright 2026 by the author(s).

## 1. Introduction

Time-series forecasting underpins a wide spectrum of real-world applications, including finance, energy, and healthcare (Torres et al., 2021). Improving predictive accuracy is crucial for decision-making, yet real world datasets often exhibit significant challenges. For example, data scarcity frequently occurs when the observation period is short, leaving models with insufficient samples to capture meaningful patterns; non-stationarity is manifested through structural breaks or regime shifts that obscure underlying regularities; and noise and anomalies, introduced by sensor failures, market manipulations, or missing values, can easily mislead forecasting models (Masini et al., 2023).

While recent deep learning models for forecasting are powerful, their effectiveness remains constrained by dataset specific meta-features (Lim & Zohren, 2021). Performance is highly conditional on the alignment between a model's design and the dataset specific meta-features (Benidis et al., 2022). For instance, Transformers (Vaswani et al., 2017) may underperform on strongly periodic data compared to Autoformer (Wu et al., 2021), while PatchTST (Nie et al., 2022) excels in univariate forecasting but may struggle in multivariate scenarios with strong intervariable dependencies. These examples highlight a fundamental limitation: each architecture is specialized for certain conditions, and no single model can generalize perfectly across all datasets (Zeng et al., 2023).

In practice, industrial applications often rely on customized frameworks that align methods like imputation and decomposition with dataset specific meta-features (Fatima & Rahimi, 2024). While this intuition is reasonable, the approach suffers from systemic limitations in large scale, dynamic environments. First, they are costly and inefficient, requiring extensive manual trial-and-error tuning by domain experts (Trirat et al., 2024). Second, they lack transferability; these brittle, fine-tuned solutions do not generalize and must be rebuilt when data conditions shift (Gong et al.). Third, they fail to adapt to data evolution; as static frameworks, their performance degrades rapidly with distribution shifts, creating an impractical burden of constant re-tuning (Fan et al., 2025). Finally, they prevent knowledge accumu-

lation, as the manual tuning process fails to distill reusable strategic insights, forcing organizations into a costly, repetitive exploration cycle for each new dataset.

These deficiencies point to a fundamental bottleneck: current practice struggles to support the demand for adaptive forecasting in complex and dynamic data environments, where systems fail to accumulate and reuse interpretable strategy knowledge and cannot achieve rapid transfer to new datasets. Recently, the rise of large language models (LLMs) has demonstrated that agents can autonomously solve complex tasks in open environments through reasoning and tool use (Guo et al., 2024). This motivates a natural question: can we build a forecasting agent that automatically learns how to select effective forecasting strategies based on dataset meta-features, thereby overcoming the inherent limitations of existing frameworks (Talagala et al., 2023)? However, training such a forecasting agent faces two central technical challenges. First, unlike NLP or vision domains, time series forecasting lacks large scale annotated corpora that map datasets to optimal pipelines, leaving agents without supervised signals for strategy learning (Das et al., 2024). Second, the relationship between dataset specific meta-features and effective forecasting strategies is highly complex and context-dependent, making it difficult to capture with handcrafted heuristics (Talagala et al., 2023). These challenges explain why no forecasting agent has yet emerged that can generalize across diverse datasets and dynamically evolving environments (Kong et al., 2025).

To address these challenges, we propose **BECRA**, a novel agent training paradigm designed to endow agents with adaptive forecasting intelligence. Unlike traditional frameworks that rely on expert heuristics and manual tuning, BECRA builds intelligence around a cycle of exploration–contrast–induction–application: it conducts structured exploration over the combinatorial space of toolchain configurations, then applies a contrastive induction operator to establish performance differences into causal strategy lessons, and encodes these lessons into an interpretable symbolic representation. In BECRA, causal lessons characterize how alternative strategy choices made by the agent influence execution outcomes under comparable dataset conditions, rather than modeling the causal mechanisms of the underlying data-generating process. This knowledge representation is not only reusable across different tasks and settings (such as zero-shot learning adaptation), but also acts as a constraint in subsequent optimization. BECRA can invoke and compose lessons through in-context planning without additional parameter updates, presenting unique capability for unprecedented sustainability in dynamic industry environment for time-series forecasting.

Our contributions can be summarized as follows: (i) We formulate adaptive time-series forecasting as an *agent training problem*, highlighting the limitations of static forecasting frameworks and pipeline recommendation strategies in dynamic, heterogeneous data environments. (ii) We propose **BECRA**, a novel agent training paradigm that enables forecasting agents to acquire transferable intelligence through contrast-aware exploration and agent-level causal lesson extraction, entirely without human-annotated supervision. (iii) We introduce a systematic strategy evaluation and selection mechanism to ensure that the retained strategy knowledge exhibits stable effectiveness and transferability, thereby enabling knowledge accumulation and zero-shot generalization to unseen datasets.

Our code is available at https://github.com/Adaptive-Forecasting-Agent/BECRA.

## 2. Related Work

**Time-Series Forecasting Models:** Classical models like ARIMA (Box & Jenkins, 1976) struggle with non-linear dependencies; although deep learning improved forecasting, modern practice is dominated by diverse Transformer variants (e.g., Informer (Zhou et al., 2021), Autoformer, PatchTST) whose performance is sensitive to dataset meta-features, so no single architecture is universally optimal. Recent time-series foundation models (TSFM) (e.g., Chronos (Ansari et al., 2024), Moirai (Woo et al., 2024)) operate as monolithic predictors whose performance primarily stems from large-scale pretraining, rather than from explicit reasoning or interpretable strategy selection. **Automated Machine Learning** methods (e.g., Auto-sklearn (Feurer et al., 2022)) can achieve strong one-off performance but rely on expensive per-dataset search and retraining.

**LLM-Agent for Time-Series Forecasting:** LLMs have enabled agents that reason and use tools (e.g., ReAct (Yao et al., 2023), AutoGPT (Yang et al., 2023), Voyager (Wang et al., 2023)). Recent work has explored LLM as agents for time-series forecasting, including augmenting forecasting models with external signals such as news (Wang et al., 2024b), or using LLMs to analyze datasets and recommend auxiliary data, preprocessing steps, or forecasting pipelines based on dataset meta-features (Yeh et al., 2025; Garza & Rosillo, 2025; Zhao et al., 2025). However, these approaches primarily position the agent as a one-shot recommender or an integration layer over predefined components. As a result, they lack a training paradigm that allows agents to accumulate transferable strategy knowledge across datasets, or to assess whether recommended strategies are actually responsible for observed performance differences. In contrast, BECRA formulates forecasting as an *agent training problem*, enabling agents to acquire reusable strategy knowledge through contrast-aware exploration and agent-level causal lesson extraction, and to generalize to unseen datasets.

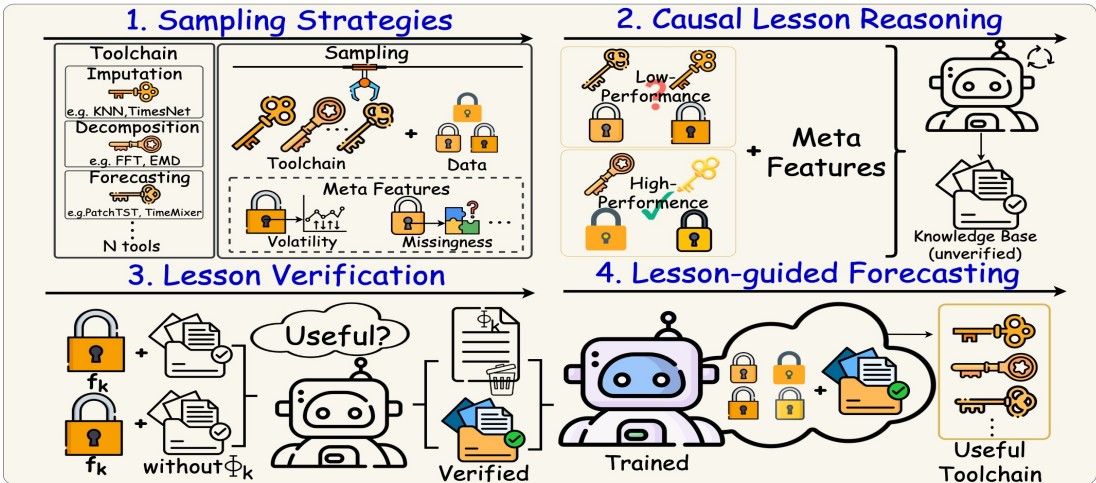

*Figure 1.* A high-level overview of the BECRA framework, illustrating the four-stage cycle.

## 3. BECRA Framework

The BECRA framework, depicted in Fig. 1, implements a four-stage cycle to endow the agent with adaptive intelligence. It begins by exploring the strategy space to build a corpus of empirical evidence, from which it induces abstract causal lessons, verifies their robustness, and ultimately uses this knowledge for zero-shot planning on unseen datasets.

### 3.1. Exploratory Construction of Forecasting Strategies

To efficiently explore the combinatorial space of time-series processing tools and forecasting models, we adopt a sampling strategy based on the Upper Confidence Bound (UCB) principle from bandit theory (Slivkins et al., 2019). Rather than focusing solely on identifying top-performing pipelines, our method, termed **Contrast-aware UCB Sampling**, is designed to collect diverse performance evidence by retaining both high-performing and low-performing strategy combinations for subsequent contrastive analysis.

Let $\mathcal{A}$ denote the set of candidate toolchains, where each $a \in \mathcal{A}$ corresponds to a predefined sequence of components (e.g., imputation method, decomposition technique, forecasting model). After evaluating a toolchain on a dataset, we update its empirical utility according to the UCB objective: $a^* = \arg\max_{a \in \mathcal{A}} \mu(a) + \lambda \cdot \sigma(a)$, where $\mu(a)$ denotes the average historical performance (e.g., inverted MSE) of toolchain $a$, $\sigma(a)$ denotes the standard deviation of its performance across datasets, reflecting uncertainty, and $\lambda$ controls the exploration–exploitation trade-off. This objective encourages sampling strategies that either exhibit strong empirical performance or high uncertainty, thereby supporting broad coverage over heterogeneous time-series characteristics.

To enable contrast-aware analysis, we explicitly retain not only high-reward pipeline–dataset pairs but also low-reward ones, particularly when they arise under similar dataset meta-feature conditions. This design ensures that the exploration stage yields balanced empirical evidence for distinguishing effective and ineffective strategies under comparable settings. The pseudocode for this procedure is provided in Appendix A.3.1.

### 3.2. Extracting Strategy Lessons via Contrastive Causal Reasoning

To generalize forecasting strategies across diverse time-series datasets, we aim to extract lessons that characterize the conditions under which particular toolchains succeed or fail. Building on the contrast-aware sampling stage, we analyze performance differences by contrasting high-performing and low-performing outcomes under comparable dataset conditions. Our focus is on attributing outcome variations to the interaction between agent strategy choices and comparable meta-feature conditions. This stage produces a set of candidate causal lessons that will be subsequently evaluated through controlled policy interventions (Section 3.3).

Formally, let $\mathcal{D} = \{(x_i, a_i, y_i)\}_{i=1}^N$ denote the sampled set, where $x_i$ represents dataset meta-features, $a_i$ denotes a forecasting strategy (i.e., a fixed toolchain), and $y_i$ is the observed performance. We partition $\mathcal{D}$ into two subsets according to a performance threshold $\tau$: $\mathcal{D}^+ = \{(x_i, a_i) \mid y_i \geq \tau\}$ and $\mathcal{D}^- = \{(x_j, a_j) \mid y_j < \tau\}$. For each strategy $a$, we construct contrastive pairs $\mathcal{C}_a = \{(x_i, x_j) \mid (x_i, a) \in \mathcal{D}^+, (x_j, a) \in \mathcal{D}^-\}$, which capture divergent outcomes of the same strategy across different datasets.

We then invoke an LLM-based agent to perform language-based induction over these contrastive pairs and distill a set of interpretable, explicitly expressible strategy lessons

$\mathcal{L} = \{\phi_1, \phi_2, \ldots, \phi_K\}$. Each lesson $\phi_k$ is a structured explanatory statement describing when a strategy is likely to be effective or ineffective. For example: $\phi_k$: *"Strategy A succeeds when the time series exhibits high volatility with frequent spikes (feature $f_k$). Because this implies the signal contains strong non-stationary jumps, which aligns with Strategy A's design that prioritizes local trend adaptation."*

Unlike conventional rule extraction methods based on supervised learning (Rudin, 2019), this process emphasizes semantic abstraction and language-based reasoning, yielding reusable and interpretable causal rules (lessons) that support generalization across datasets. The pseudocode for this stage is provided in Appendix A.3.2.

### 3.3. Lesson Verification via Controlled Policy Interventions

The strategy lessons induced in the previous stage are hypothesis-level descriptions and may not be reliable under distributional variation. We therefore introduce a verification stage to assess whether a lesson consistently influences execution outcomes when used as a planning prior. Importantly, the notion of causality adopted in BECRA is defined at the level of agent decision-making. Following the interventionist account of causation (Woodward & Woodward, 2005), a lesson is considered valid if intervening on the agent's strategy selection policy leads to a consistent outcome difference under fixed dataset conditions. Each lesson $\phi_k$ is treated as a symbolic hypothesis (e.g., "strategy $A$ is effective when feature pattern $f_k$ holds"). To evaluate it, we measure the causal effect: $\Delta_{\phi_k} = P(y = 1 \mid f_k, \phi_k) - P(y = 1 \mid f_k, \neg\phi_k)$, where $y = 1$ denotes a successful outcome. This quantity captures the causal effect of enforcing $\phi_k$ as a planning prior at the level of agent decision-making, measured under matched execution conditions that fix the dataset, toolchain, and decoding configuration.

We estimate $\Delta_{\phi_k}$ using paired rollouts under controlled execution settings. Specifically, the agent performs two planning episodes on the same dataset with identical tool library, prompts, and decoding configurations, differing only in whether $\phi_k$ is injected as a planning constraint. This paired comparison constitutes a controlled intervention on the agent's decision policy, enabling causal assessment of whether enforcing the lesson induces a consistent directional change in outcomes across stochastic planning executions.

To ensure robustness, we apply the same verification procedure to both positive and negative lessons. For a negative lesson $\phi^-$ (e.g., "avoid strategy $S$"), we compare performance when the agent follows $\phi^-$ versus when it is forced to contradict it. The corresponding outcome difference is defined as $\Delta_{\phi^-} = P(y = 1 \mid f_k, \phi^-) - P(y = 1 \mid f_k, \text{Contradict}(\phi^-))$. A consistently positive $\Delta_{\phi^-}$ indicates

that adhering to the negative lesson yields better outcomes than violating it.

A lesson is retained in the knowledge base only if its estimated outcome difference exceeds a predefined confidence threshold; otherwise, it is discarded as low-confidence. We define success using a relative criterion: $y = 1$ if a strategy's performance ranks within the top $\alpha\%$ on the dataset, and $y = 0$ otherwise. This filtering step ensures that only empirically supported and intervention-validated lessons are used for downstream planning. The pseudocode for this verification procedure is provided in Appendix A.3.3.

### 3.4. Forecasting with Lesson-Guided Planning

With a library of validated symbolic lessons, BECRA transitions from exploration to deployment. Rather than relying on fine-tuning, forecasting is performed via in-context learning (ICL) (Dong et al., 2022), where each lesson serves as a symbolic planning prior. Given a new dataset characterized by meta-features $x_{\text{new}}$, the agent retrieves relevant lessons $\mathcal{L}_x \subset \mathcal{L}$ whose conditions match $x_{\text{new}}$, and injects them as structured prompts to guide planning over forecasting toolchains. As a result, decisions are informed by accumulated causal knowledge at the level of agent decision-making, rather than memorized statistical associations.

We choose in-context learning over fine-tuning for three reasons. First, time-series datasets are often small and heterogeneous, making fine-tuning prone to learning dataset-specific correlations rather than transferable causal structure, which results in fragile performance under distribution shifts (Yin et al., 2024). In contrast, ICL leverages symbolic induction to distill reusable strategy knowledge that generalizes across domains. Second, fine-tuning is computationally expensive and inflexible in dynamic environments (Mosbach et al., 2023), as frequent retraining is required to cope with evolving data distributions. BECRA instead enables zero-training adaptation through lesson memory, allowing instantaneous strategy planning on new datasets. Finally, ICL allows BECRA to exploit powerful closed-source LLMs that cannot be fine-tuned, while still steering their reasoning via symbolic lessons. This design supports efficient and scalable adaptation without any model retraining. The pseudocode for this stage is provided in Appendix A.3.4.

### 3.5. Meta-Feature and Toolchain Design in BECRA

In BECRA, each dataset is represented by a set of scalar meta-features that provide a compact, quantitative interface for agent decision-making. Following common practice in time-series meta-learning (Lemke & Gabrys, 2010; Wei et al., 2025; Talagala et al., 2023; Zhao et al., 2025), these features are organized into functional groups aligned with key modeling needs (e.g., missingness, periodicity, stationarity), together with complementary characteristics

*Table 1.* Long-term forecasting results. All results are averaged over four prediction lengths, i.e., {96, 192, 336, 720}, while the input sequence length is fixed at 96.

| Models Metric | BECRA | | ChatGPT-4 | | Gemini | | LLaMA | | TimeMixer | | TimesNet | | PatchTST | | iTransformer | | TimeLLM | | GPT4TS | | Chronos | | Moirai | |
|---|---|---|---|---|---|---|---|---|---|---|---|---|---|---|---|---|---|---|---|---|---|---|---|---|
| | MSE | MAE | MSE | MAE | MSE | MAE | MSE | MAE | MSE | MAE | MSE | MAE | MSE | MAE | MSE | MAE | MSE | MAE | MSE | MAE | MSE | MAE | MSE | MAE |
| ETTh1 | **0.441** | **0.436** | 0.472 | 0.479 | 0.452 | 0.446 | 0.566 | 0.520 | 0.467 | 0.450 | 0.479 | 0.477 | 0.451 | 0.441 | 0.464 | 0.455 | 0.460 | 0.490 | 0.447 | 0.436 | 0.588 | 0.466 | 0.480 | 0.439 |
| ETTh2 | **0.367** | **0.375** | 0.384 | 0.409 | 0.486 | 0.466 | 0.598 | 0.544 | 0.432 | 0.432 | 0.446 | 0.451 | 0.413 | 0.420 | 0.428 | 0.434 | 0.389 | 0.408 | 0.381 | 0.408 | 0.455 | 0.427 | 0.367 | 0.377 |
| ETTm1 | **0.380** | **0.389** | 0.768 | 0.597 | 0.875 | 0.640 | 1.288 | 0.703 | 0.3841 | 0.397 | 0.408 | 0.417 | 0.383 | 0.395 | 0.407 | 0.412 | 0.395 | 0.390 | 0.389 | 0.397 | 0.555 | 0.465 | 0.422 | 0.391 |
| ETTm2 | **0.274** | **0.321** | 1.058 | 0.720 | 1.267 | 0.803 | 1.245 | 0.795 | 0.278 | 0.325 | 0.295 | 0.332 | 0.285 | 0.327 | 0.291 | 0.334 | 0.281 | 0.321 | 0.285 | 0.331 | 0.295 | 0.338 | 0.329 | 0.343 |
| Electricity | **0.171** | **0.264** | 0.211 | 0.293 | 0.198 | 0.293 | 0.202 | 0.300 | 0.185 | 0.275 | 0.194 | 0.296 | 0.190 | 0.275 | 0.180 | 0.270 | 0.175 | 0.265 | 0.205 | 0.290 | 0.204 | 0.273 | 0.186 | 0.270 |
| Weather | **0.240** | **0.270** | 0.258 | 0.281 | 0.278 | 0.302 | 0.262 | 0.283 | 0.244 | 0.275 | 0.251 | 0.288 | 0.258 | 0.280 | 0.335 | 0.363 | 0.250 | 0.274 | 0.274 | 0.290 | 0.279 | 0.306 | 0.264 | 0.273 |

such as volatility and trend structure. Crucially, BECRA does not aim to reconstruct raw time series from meta-features; instead, their primary requirement is discriminative power, enabling a stable mapping from dataset characteristics to effective toolchain components. This design involves a natural trade-off: overly sparse meta-features may under-specify tool applicability, while overly engineered or domain-specific descriptors can introduce noise and reduce interpretability and cross-domain transfer. BECRA addresses this risk through the lesson verification stage (Section 3.3), which filters out low-confidence lessons that do not exhibit stable performance associations under the corresponding meta-feature conditions. A full list of meta-features is provided in Appendix A.1.

BECRA operates on a modular six-stage forecasting pipeline that includes imputation, anomaly handling, transformation, decomposition, normalization, and forecasting, following conventional system-level design paradigms in the time-series forecasting literature (Montgomery et al., 2015; Liu et al., 2021). Rather than executing all stages by default, BECRA dynamically selects only the necessary processing stages and corresponding tools for each dataset. The construction of the tool library follows two guiding principles: *(i) diversity of inductive biases* and *(ii) modular coverage*. Diversity is critical because BECRA acquires causal strategy knowledge through contrastive comparisons—if the tool library is homogeneous, informative contrasts collapse and weaken the success/failure evidence required for strategy induction. Modular coverage ensures that the tool library spans the entire forecasting pipeline, enabling different data pathologies characterized by meta-features to be mapped to appropriate candidate tool spaces. The complete tool library is provided in Appendix A.2. Further analysis about the roles of meta-features and toolchain diversity in Appendix A.5.

## 4. Experiments

### 4.1. Experimental Setup

Our experiments are conducted on a diverse set of widely-used benchmarks, including ETTh1, ETTh2, ETTm1,

ETTm2 (Zhou et al., 2021), Electricity (Lee et al., 2018), Weather (Godahewa et al., 2021), and the large-scale, highly heterogeneous M4 dataset (Makridakis et al., 2020). We evaluate BECRA against three categories of baselines: untrained agents powered by LLM (e.g., ChatGPT-4 (Achiam et al., 2023), Gemini (Team et al., 2023), and LLaMA (Touvron et al., 2023)), SOTA single-architecture models (TimeMixer (Wang et al., 2024a), TimesNet (Wu et al., 2022), PatchTST, iTransformer (Liu et al., 2023)), LLM-based models (e.g., TimeLLM (Jin et al., 2023) & GPT4TS (Zhou et al., 2023)) and pre-trained time-series foundation models (Chronos, Moirai). To ensure a fair comparison, we distinguish between the evaluation protocols. Our proposed BECRA agent, powered by ChatGPT-4, is evaluated in a zero-shot training manner: it leverages strategic lessons learned exclusively from other datasets to construct a forecasting framework for the target dataset, without training on it thereby ensuring the evaluation rigorously tests the agent's ability to generalize to a truly unseen dataset, rather than its ability to recall a known optimal solution. Similarly, the untrained LLM agent baselines are evaluated on their strategic capabilities; they are given the target dataset's meta-features and tasked with recommending a pipeline for prediction. In contrast, all other baselines (single-architecture and foundation models) follow the standard protocol of being trained or fine-tuned on the target dataset's training split. This design allows us to fairly assess both BECRA's strategic intelligence against other agents and its final forecasting accuracy against state-of-the-art forecasters. The prompt templates used are provided in Appendix A.4.

All experiments are conducted on a cluster with dual NVIDIA 5090 GPUs. For consistency, the sequence length is fixed at 96 for all experiments. We evaluate long-term forecasting on prediction lengths 96, 192, 336, 720 and short-term forecasting on the M4 dataset with input lengths [12, 96] and prediction lengths [6, 48], respectively. We use standard metrics like MSE, MAE, SMAPE, MASE, and OWA to measure performance. Detailed experimental information is in Appendix A.9.

*Table 2.* Short-term forecasting results on the M4 dataset. The input length and prediction length are set to [12, 96] and [6, 48], respectively.

| | Models | BECRA | TimeLLM | GPT4TS | PatchTST | FEDformer | Autoformer | TimesNet | ChatGPT-4 | Gemini | LLaMA |
|---|---|---|---|---|---|---|---|---|---|---|---|
| Yearly | SMAPE | **13.275** | 13.419 | 13.531 | 13.477 | 13.728 | 13.974 | 13.387 | 14.920 | 13.418 | 13.436 |
| | MASE | **2.984** | 3.005 | 3.015 | 3.019 | 3.048 | 3.134 | 2.996 | 3.364 | 3.045 | 3.043 |
| | OWA | **0.782** | 0.789 | 0.793 | 0.792 | 0.803 | 0.822 | 0.786 | 0.880 | 0.793 | 0.794 |
| Quarterly | SMAPE | **9.997** | 10.100 | 10.177 | 10.380 | 10.792 | 11.338 | 10.100 | 11.122 | 10.202 | 10.124 |
| | MASE | **1.166** | 1.178 | 1.194 | 1.233 | 1.283 | 1.365 | 1.182 | 1.360 | 1.194 | 1.169 |
| | OWA | **0.879** | 0.889 | 0.898 | 0.921 | 0.958 | 1.012 | 0.890 | 1.001 | 0.899 | 0.886 |
| Monthly | SMAPE | **12.633** | 12.980 | 12.894 | 12.959 | 14.260 | 13.958 | 12.679 | 15.626 | 12.791 | 12.677 |
| | MASE | **0.922** | 0.963 | 0.956 | 0.970 | 1.102 | 1.103 | 0.933 | 1.274 | 0.969 | 0.937 |
| | OWA | **0.871** | 0.903 | 0.897 | 0.905 | 1.012 | 1.002 | 0.878 | 1.141 | 0.899 | 0.880 |
| Others | SMAPE | **4.696** | 4.795 | 4.940 | 4.952 | 4.954 | 5.485 | 4.891 | 7.186 | 5.061 | 4.925 |
| | MASE | **3.189** | 3.178 | 3.228 | 3.347 | 3.264 | 3.865 | 3.302 | 4.677 | 3.216 | 3.391 |
| | OWA | **0.997** | 1.006 | 1.029 | 1.049 | 1.036 | 1.187 | 1.035 | 1.494 | 1.040 | 1.053 |

## 4.2. Long-Term Forecasting Results

We conducted a comprehensive evaluation of BECRA's long-term forecasting performance across six benchmark datasets, with the averaged results presented in Table 1. Our evaluation on BECRA employs a strict leave-one-out protocol: for each target dataset, the agent's knowledge base is constructed from causal lessons extracted exclusively from the other datasets. This ensures the agent has zero prior knowledge of the target's specific strategy to performance mapping, testing its ability to generalize rather than recall. The results in Table 1 validate that BECRA can leverage its distilled symbolic lessons via in-context planning to select and compose a high-performing forecasting toolchain for entirely unseen datasets, achieving true **zero training adaptation**. Furthermore, the results confirm BECRA's **cross-domain transferability**. For instance, despite fundamental differences between datasets like Electricity and Weather, the agent consistently generated optimal forecasting toolchains for a target domain using lessons sourced from entirely different ones. This demonstrates that by extracting abstract, domain-agnostic causal relationships between meta-features and strategies, BECRA shows the ability to transfer intelligence across domains.

Evaluation against three key baseline categories demonstrates the multifaceted advantages of the BECRA paradigm. First, its performance surpasses untrained LLM-powered agents (e.g., ChatGPT-4, Gemini, LLaMA), highlighting the necessity of the BECRA training paradigm. The results indicate that, while LLMs exhibit powerful general-purpose reasoning, such capabilities are not innately suited for time-series forecasting. A specialized, unsupervised process of distilling causal lessons, such as BECRA's, is essential to transform them into domain-expert planners. Second, BECRA's adaptive agent design consistently outperforms specialized single-architecture models (e.g., TimeMixer, TimesNet, PatchTST, iTransformer). Unlike these models, which are bound by a fixed inductive bias, BECRA learns a higher-order strategy of aligning the appropriate bias with dataset meta-features. Finally, we compare BECRA against emerging LLM-based (e.g., TimeLLM & GPT4TS) and pre-trained time-series foundation models (e.g., Chronos and Moirai). As shown in Table 1, BECRA demonstrates consistently superior performance. Its ability to outperform large-scale models, while relying on a relatively lightweight toolchain, suggests that strategic intelligence, which is the knowledge of how to decide, is more important than raw model capacity or the breadth of available tools. This finding is further reinforced by our targeted experimental design. We deliberately excluded time-series foundation models from the BECRA's tool library. This was motivated by two primary considerations: (1) to prevent the agent from learning a 'shortcut' of simply relying on a powerful model's capacity instead of engaging in the fine-grained analysis of meta-features and algorithmic principles, which would compromise our research objectives; (2) to mitigate the risk of data leakage from foundation models that may have been exposed to evaluation datasets. Uncertainty estimation results and statistical analysis are provided in Appendix A.7.

## 4.3. Short-Term Forecasting Results

We evaluated BECRA on the large-scale and highly heterogeneous M4 dataset for short-term forecasting, with the results presented in Table 2. The experiment shows that BECRA achieves best performance across all frequency categories (Yearly, Quarterly, Monthly, etc.), comprehensively outperforming the baseline models. This result provides evidence for the cross-task generalization capability of the strategic lessons distilled by BECRA. It demonstrates that the abstract causal knowledge learned from long-term forecasting tasks can be successfully transferred and adaptively applied to short-term problems across diverse frequencies and domains in a zero-shot manner.

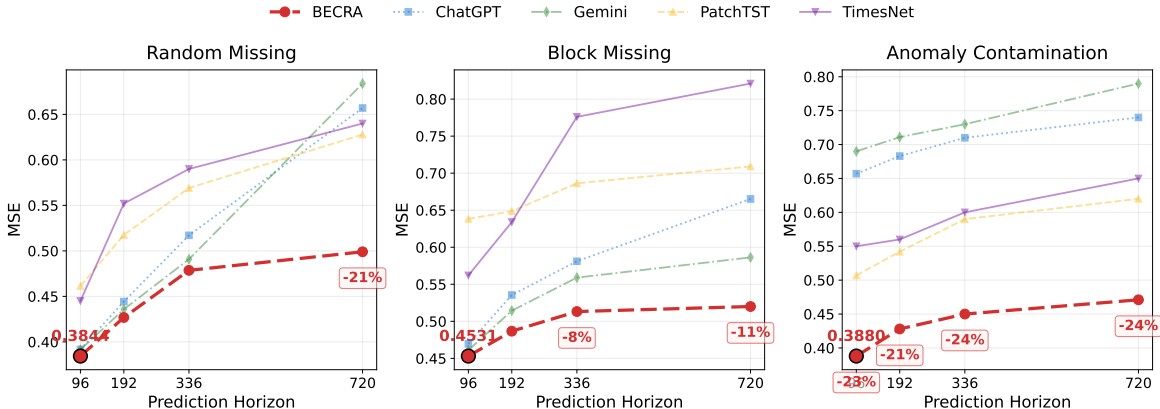

*Figure 2.* **BECRA Robustness Evaluation under Data Missing and Anomaly Scenarios on the ETTh1 dataset.** The plots show MSE as the prediction horizon increases, with results averaged over several severity levels. Left: Random Missing (averaged over 15%, 25%, 35% missing rates). Middle: Block Missing (averaged over 15%, 25%, 35% missing rates). Right: Anomaly Contamination (averaged over 2%, 5%, 10% rates).

## 4.4. Robustness Evaluation under Data Missing and Anomaly Contamination Scenarios

To evaluate robustness under imperfect data, we conduct stress tests on the ETTh1 dataset by introducing missing values with two representative patterns: *Random Missing*, simulating sporadic issues such as sensor noise or packet loss, and *Block Missing*, modeling systematic interruptions such as batch failures. We progressively increase the missing ratio (15%, 25%, 35%) and extend the prediction horizon from 96 to 720 steps to assess BECRA under increasingly challenging conditions. As shown in Fig. 2 (left and middle subplots), BECRA consistently achieves the lowest MSE across both missing patterns. More importantly, it exhibits strong robustness: its error curve remains substantially flatter than those of the baselines, indicating significantly more graceful degradation as the prediction horizon increases. This demonstrates BECRA's stability under the combined challenges of long-range forecasting and incomplete data.

We further assess robustness by injecting random outliers into the ETTh1 dataset at contamination ratios of 2%, 5%, and 10%. This stress test evaluates whether BECRA can adapt its strategy to mitigate anomaly interference under long-range forecasting. As shown in Fig. 2 (right subplot), BECRA consistently maintains the lowest MSE on contaminated data. Moreover, its performance remains highly stable, with an almost flat error curve as the prediction horizon increases, in contrast to the steadily degrading baselines. The results confirm BECRA's robustness against the combined effects of anomalies and extended forecasting horizons.

## 5. Ablation Study

To deconstruct our paradigm and validate its core components, we conducted four targeted ablation studies. For a

fair comparison with our main findings, these experiments used the identical datasets, toolchains, evaluation protocols, and prediction horizons as the long-term forecasting tasks.

First, to validate the necessity of our causal lessons, we ablate the entire lesson-guided planning module in **w/o Lessons**. This variant operates directly on the experience database constructed by our Contrast-aware UCB Sampling, which contains both high- and low-performing toolchain combinations. It therefore knows what strategies performed well or poorly on certain datasets, but lacks the abstracted symbolic lessons that explain why. As shown in Table 3, the performance of this variant collapses. This provides evidence that these symbolic lessons are the core of the agent's intelligence; the deep, causal understanding they provide is the foundational reason for our framework's ability to generalize effectively to new datasets.

Second, having established the critical value of lessons, we investigate the optimal method for an agent to leverage them. This ablation validates our choice of using in-context learning (ICL), where lessons act as dynamic, zero-shot prompts, against the alternative of using them as training data for supervised fine-tuning (SFT). For the SFT variant, we fine-tuned three representative **LLMs** to map meta-features and lessons to the optimal toolchain. As shown in Table 3, our ICL-based agent outperforms the SFT counterparts. We attribute SFT's underperformance to two primary issues: 1) its tendency to overfit on limited training data by memorizing specific pairs instead of internalizing general principles, and 2) the risk of damaging the LLM's powerful pre-trained reasoning abilities. Our ICL approach avoids these pitfalls and offers higher computational efficiency (as it requires no costly retraining) and scalability (new lessons can be instantly integrated into the knowledge database). This makes ICL far better suited for the dynamic, continuously learning

*Table 3.* Ablation study of BECRA components. **BECRA**: Full components. **Greedy**: Uses greedy sampling instead of contrast-aware UCB. **w/o Lessons**: Removes lesson-guided planning. **w/o Verification**: Skips the lesson verification stage. **Finetuned**: Agents using supervised fine-tuning instead of in-context learning. All results are averaged over four prediction lengths, i.e., {96, 192, 336, 720}, while the input sequence length is fixed at 96. The LLaMA-7B, ChatGLM-6B (GLM et al., 2024) and Mistral-7B (Jiang et al., 2023) baselines are based on relatively small-scale models.

| Dataset | BECRA | | Greedy | | w/o Lessons | | w/o Verification | | LLaMA (Finetuned) | | ChatGLM (Finetuned) | | Mistral (Finetuned) | |
|---|---|---|---|---|---|---|---|---|---|---|---|---|---|---|
| | MSE | MAE | MSE | MAE | MSE | MAE | MSE | MAE | MSE | MAE | MSE | MAE | MSE | MAE |
| ETTh1 | **0.441** | **0.436** | 0.450 | 0.445 | 0.563 | 0.520 | 0.465 | 0.450 | 0.443 | 0.439 | 0.442 | 0.439 | 0.442 | 0.438 |
| ETTh2 | **0.367** | **0.375** | 0.412 | 0.420 | 0.587 | 0.534 | 0.432 | 0.431 | 0.368 | 0.397 | 0.395 | 0.401 | 0.369 | 0.399 |
| ETTm1 | **0.380** | **0.389** | 0.384 | 0.396 | 1.197 | 0.712 | 0.384 | 0.397 | 0.384 | 0.397 | 0.392 | 0.399 | 0.383 | 0.394 |
| ETTm2 | **0.274** | **0.321** | 0.284 | 0.326 | 1.034 | 0.722 | 0.277 | 0.326 | 0.281 | 0.327 | 0.285 | 0.324 | 0.276 | 0.322 |
| Electricity | **0.171** | **0.264** | 0.191 | 0.275 | 0.203 | 0.294 | 0.173 | 0.272 | 0.184 | 0.274 | 0.191 | 0.278 | 0.186 | 0.275 |
| Weather | **0.240** | **0.270** | 0.257 | 0.281 | 0.258 | 0.281 | 0.243 | 0.271 | 0.256 | 0.280 | 0.255 | 0.281 | 0.258 | 0.283 |

environments found in real-world applications.

Third, we test the importance of the Lesson Verification stage. In this ablation **w/o Verification**, the agent uses all raw lessons directly from the induction phase without any filtering. The results show a consistent, albeit slight, performance degradation. This drop is attributed to spurious or flawed lessons that act as noise and can mislead the agent's planning. The verification stage thus serves as a crucial quality-control filter, pruning these unreliable priors to ensure the agent's final knowledge base is robust, generalizable, and reliable.

Finally, we validate our Contrast-aware UCB Sampling strategy against a **Greedy** variant that collects only high-performing (positive) examples. The agent relying on lessons from this success-only data performs significantly worse. This is because a Greedy approach lacks the necessary contrast for deep causal reasoning. While it might learn shallow correlations (what works), it cannot understand the crucial context of why a strategy is effective. By analyzing both successes and failures, our contrast-aware method enables the distillation of deeper, causal lessons, confirming that contrast is a prerequisite for extracting profound and actionable knowledge.

## 6. Transferability and Efficiency

To evaluate the generalizability and model-agnostic nature of the BECRA training paradigm, we conducted a transferability analysis. Keeping the knowledge base (lesson memory) and prompt structure identical, we replaced the agent's original reasoning engine with a variety of state-of-the-art closed and open-source LLMs. These experiments used the identical datasets, toolchains, evaluation protocols, and prediction horizons as the long-term forecasting tasks.

The results in Table 4 show comparable performance across all tested LLMs. While the original agent performs best, other powerful models like Claude 3.5 show only a marginal and graceful degradation. This is an important finding: it demonstrates that the strategic guidance within the symbolic

*Table 4.* Transferability analysis of the BECRA paradigm across different LLMs. All the results are averaged from 4 different prediction lengths, that is {96, 192, 336, 720} while sequence length is fixed at 96. All LLM baselines are accessed via API: Claude 3.5 (Anthropic, 2024), Gemini Pro (Team et al., 2023), Qwen-2.5-72B (Yang et al., 2025), and LLaMA-3.1-70B (Grattafiori et al., 2024).

| Dataset | BECRA | | Claude 3.5 | | Gemini Pro | | Llama-3.1 70B | | Qwen-2.5 72B | |
|---|---|---|---|---|---|---|---|---|---|---|
| | MSE | MAE | MSE | MAE | MSE | MAE | MSE | MAE | MSE | MAE |
| ETTh1 | **0.441** | **0.436** | 0.442 | 0.438 | 0.445 | 0.441 | 0.446 | 0.442 | 0.444 | 0.440 |
| ETTh2 | **0.367** | **0.375** | 0.369 | 0.397 | 0.372 | 0.400 | 0.375 | 0.402 | 0.370 | 0.397 |
| ETTm1 | **0.380** | **0.389** | 0.382 | 0.392 | 0.385 | 0.394 | 0.388 | 0.397 | 0.383 | 0.392 |
| ETTm2 | **0.274** | **0.321** | 0.276 | 0.323 | 0.279 | 0.325 | 0.282 | 0.328 | 0.277 | 0.323 |
| Electricity | **0.171** | **0.264** | 0.173 | 0.272 | 0.176 | 0.274 | 0.182 | 0.278 | 0.174 | 0.272 |
| Weather | 0.240 | 0.270 | **0.239** | **0.270** | 0.243 | 0.273 | 0.254 | 0.281 | 0.247 | 0.276 |

lessons distilled by the BECRA paradigm is portable and can be successfully understood by advanced LLMs. This successful decoupling of the "knowledge" from the "reasoner" is significant as it validates the generalizability of BECRA itself. Ultimately, this confirms BECRA is a robust paradigm for creating portable forecasting intelligence, not just a model-specific solution.

In addition, BECRA can also significantly improve efficiency. Compared with representative AutoML baselines, BECRA has reduced trial costs by more than 80% through its transferable lesson memory and sampling parameters. Details of **computational cost analysis** is in Appendix A.6.

## 7. Conclusion

We propose BECRA, the first agent training paradigm for time-series forecasting that integrates contrast-aware UCB sampling with lesson-guided planning to construct reusable strategy memory. BECRA addresses a fundamental challenge in time-series forecasting: the lack of scalable supervision for learning transferable mappings from heterogeneous data characteristics to effective forecasting strategies across domains. Extensive experiments demonstrate that BECRA achieves strong performance, robustness, and efficiency,

while remaining portable across large language models. Beyond time series, the exploration–contrast–induction mechanism of BECRA offers a general paradigm for knowledge acquisition in data-scarce domains where matching data with effective tools is challenging, such as materials science and drug discovery. By replacing expensive per-task searches with reusable strategic reasoning, BECRA provides a scalable path for transforming expert domain knowledge into practical learning frameworks. Further discussion in Appendix A.8.

## Acknowledgements

**Funding Disclosure.** This research received no specific grant from any funding agency in the public, commercial, or not-for-profit sectors. Carl Yang was not supported by any fund from China.

**Author Contributions.** Qingwen Zeng, Dajun Guo, and Zhaoge Bi contributed equally to this work. Qingwen Zeng contributed to the conceptualization, methodological design, and manuscript writing. Dajun Guo contributed to method development and experiment implementation. Zhaoge Bi contributed to experiment implementation and experimental analysis. Lining Chen contributed to experiment implementation and experimental analysis. Jushang Qiu contributed to visualization and manuscript writing. Yitian Yang contributed to visualization and discussions. Carl Yang, Huaming Chen, and Ling Chen supervised the project and provided senior guidance and feedback. Huaming Chen served as the corresponding author.

## Impact Statement

This paper presents a novel solution to reduce the manual effort of building and maintaining time-series forecasting pipelines by learning transferable strategy knowledge. We expect this work could improve forecasting reliability in numerous critical domains, supporting better planning and resource allocation. It is suggested that the method will be used as decision support with careful monitoring and evaluation prior to deployment, especially in safety-critical scenarios.

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

# A. Appendix

## A.1. Meta-Feature Descriptions

This appendix provides a detailed description of the meta-features used by the BECRA agent to characterize time-series datasets. For each feature, we describe its significance for forecasting strategy selection and the specific metrics used for its quantification.

### A.1.1. BASIC INFORMATION

This group of features captures the fundamental scale and dimensionality of the dataset.

**Series Length**    The total number of observations (time steps) in the series. This metric informs the agent about the volume of historical data available. Very short series may be insufficient for training complex models, making simpler models or transfer learning more appropriate.

**Number of Variates**    The number of parallel time series variables in the dataset. A value of 1 indicates a univariate task, while a value greater than 1 indicates a multivariate task. This is a critical feature for deciding whether to use models that can capture cross-channel dependencies (e.g., TimesNet) or channel-independent models (e.g., PatchTST).

### A.1.2. MISSINGNESS

This feature quantifies the extent of missing data, which is a common problem in real-world time series.

**Missing Rate**    This metric measures the proportion of missing ('NaN') values in the dataset. It is calculated as:

$$\text{Missing Rate} = \frac{\text{Count of NaN values}}{\text{Total number of observations}}$$

A high missing rate signals that an imputation strategy is essential. The agent uses this value to decide whether to apply imputation and to select an appropriate method (e.g., simple interpolation for low rates vs. a sophisticated model like TimesNet for high rates).

### A.1.3. VOLATILITY

Volatility features measure the degree of fluctuation or dispersion in the series.

**Variance** ($\sigma^2$)    Variance measures the average squared deviation of each observation from the series mean ($\mu$), indicating the overall magnitude of fluctuations. It is calculated as:

$$\sigma^2 = \frac{1}{N} \sum_{i=1}^{N} (x_i - \mu)^2$$

where $N$ is the series length and $x_i$ is the value at time $i$.

**Standard Deviation** ($\sigma$)    As the square root of the variance, the standard deviation is expressed in the same units as the data, making it more interpretable.

$$\sigma = \sqrt{\frac{1}{N} \sum_{i=1}^{N} (x_i - \mu)^2}$$

High variance or standard deviation suggests a highly fluctuating series, which may benefit from smoothing, decomposition, or models designed to handle volatility.

**Coefficient of Variation (CV)**    The CV measures relative volatility by normalizing the standard deviation by the mean.

$$\text{CV} = \frac{\sigma}{\mu}$$

This is particularly useful for comparing the volatility of series with different average values. A high CV indicates significant volatility relative to the series' scale.

A.1.4. TREND STRUCTURE

These features assess the presence and nature of trends in the data.

**Linear and Non-linear Trend Scores** We assess trend by fitting regression models to the time series, with time as the independent variable. The "score" is the coefficient of determination ($R^2$), which measures the proportion of the variance in the series that is predictable from the time variable. For a linear trend, the model is $y_t = \beta_0 + \beta_1 t + \epsilon_t$. For a non-linear trend, a polynomial model such as $y_t = \beta_0 + \beta_1 t + \beta_2 t^2 + \epsilon_t$ can be used. The $R^2$ is calculated as:

$$R^2 = 1 - \frac{\sum_{i=1}^{N}(y_i - \hat{y}_i)^2}{\sum_{i=1}^{N}(y_i - \bar{y})^2}$$

where $y_i$ is the observed value, $\hat{y}_i$ is the value predicted by the regression model, and $\bar{y}$ is the mean of the series. A high $R^2$ value indicates a strong trend, guiding the agent to include trend modeling or detrending components in the toolchain.

A.1.5. PERIODICITY

Periodicity features are crucial for identifying recurring cyclical patterns (seasonality).

**Autocorrelation Function (ACF)** The ACF measures the correlation of the time series with a lagged version of itself. The value at lag $k$ is calculated as:

$$\text{ACF}(k) = \frac{\sum_{t=k+1}^{N}(x_t - \mu)(x_{t-k} - \mu)}{\sum_{t=1}^{N}(x_t - \mu)^2}$$

Strong, significant peaks in the ACF plot at regular intervals are a clear sign of seasonality. The agent uses the magnitude of these peaks to gauge the strength of periodicity.

**Partial Autocorrelation Function (PACF)** The PACF measures the correlation between an observation at time $t$ and an observation at time $t - k$, after removing the linear effects of the intermediate lags ($t - 1, t - 2, \ldots, t - k + 1$). It helps to identify the direct relationship between observations. Significant PACF values help in determining the order of autoregressive models.

**Fourier Spectral Peaks** By applying a Fast Fourier Transform (FFT) to the series, we can decompose it into a sum of sine and cosine waves of different frequencies. The resulting power spectrum shows the magnitude of each frequency. Prominent peaks in the spectrum correspond to the dominant periodicities in the data. The agent uses the location and height of these peaks to confirm seasonality and its period, guiding the choice of frequency-based models (e.g., FEDformer) or seasonal decomposition.

A.1.6. REGIME SHIFT

This feature identifies abrupt and fundamental changes in the statistical properties of the series.

**Change Points** A change point is a time index where the statistical properties of the series (e.g., mean, variance, trend) change significantly. We identify these points using statistical algorithms like the Pruned Exact Linear Time (PELT) method. The metric is the total count of significant change points detected. A high number of change points suggests structural breaks, which may warrant the use of segment-wise modeling or adaptive forecasting methods.

A.1.7. STATIONARITY

Stationarity implies that the statistical properties of a series do not change over time. Many forecasting models assume stationarity.

**Augmented Dickey-Fuller (ADF) Test** The ADF test is a statistical test for a unit root in a time series sample. Its null hypothesis is that a unit root is present (the series is non-stationary). The test is based on the regression model:

$$\Delta y_t = \alpha + \beta t + \gamma y_{t-1} + \delta_1 \Delta y_{t-1} + \cdots + \delta_{p-1} \Delta y_{t-p+1} + \epsilon_t$$

The test statistic is used to compute a p-value for the coefficient $\gamma$. A low p-value (e.g., $< 0.05$) leads to the rejection of the null hypothesis, suggesting the series is stationary.

**KPSS Test**   The Kwiatkowski-Phillips-Schmidt-Shin (KPSS) test has a null hypothesis that the series is stationary around a deterministic trend. It is often used as a confirmatory test alongside ADF. A low p-value (e.g., $< 0.05$) indicates that the null hypothesis is rejected and the series is non-stationary. The agent uses the results of both tests to make a robust decision on whether differencing or other transformations are required.

A.1.8. DISTRIBUTIONAL SHAPE

These features describe the shape of the data's probability distribution, providing insights into non-Gaussian properties.

**Skewness**   Skewness measures the asymmetry of the distribution. A value of 0 indicates a perfectly symmetric distribution. The sample skewness is calculated as:

$$g_1 = \frac{\frac{1}{N} \sum_{i=1}^{N} (x_i - \mu)^3}{(\sigma^2)^{3/2}}$$

Significant skewness may violate the assumptions of some models and suggests that a data transformation (e.g., a log or Box-Cox transform) could be beneficial.

**Kurtosis**   Kurtosis measures the "tailedness" of the distribution compared to a normal distribution. Excess kurtosis (kurtosis minus 3) is often used, where a value greater than 0 indicates heavy tails and a higher likelihood of outliers. The sample excess kurtosis is:

$$g_2 = \frac{\frac{1}{N} \sum_{i=1}^{N} (x_i - \mu)^4}{(\sigma^2)^2} - 3$$

High kurtosis can guide the agent to choose models or loss functions that are more robust to outliers.

A.1.9. NOISE CHARACTERISTICS

These features quantify the level of random, unpredictable noise in the series.

**Signal-to-Noise Ratio (SNR)**   The SNR compares the level of the desired signal to the level of background noise. It is often expressed in decibels (dB):

$$\text{SNR}_{\text{dB}} = 10 \log_{10} \left( \frac{P_{\text{signal}}}{P_{\text{noise}}} \right) = 20 \log_{10} \left( \frac{\sigma_{\text{signal}}}{\sigma_{\text{noise}}} \right)$$

where $P$ is power and $\sigma$ is the standard deviation. The signal and noise can be separated using filters or smoothing (where the smoothed series is the signal and the residual is the noise). A low SNR indicates a noisy series where denoising could be a valuable preprocessing step.

**Entropy**   Shannon entropy measures the amount of uncertainty or randomness in the data. For a time series, we can bin the values to create a probability distribution and calculate entropy as:

$$H(X) = - \sum_{i=1}^{n} p(x_i) \log_2 p(x_i)$$

where $p(x_i)$ is the probability of the series taking a value in the $i$-th bin. A high entropy value suggests the series is highly unpredictable and may be difficult to forecast accurately.

**A.2. Comprehensive Tool Library**

This appendix provides a comprehensive description of the tools available to the BECRA agent for each stage of the time-series processing pipeline. The diversity of these tools enables the agent to compose a wide variety of strategies tailored to different dataset characteristics.

## 1. IMPUTATION

- **Mean / Median / Mode**: Simple statistical imputation, fast but may distort variance.

- **Forward / Backward Fill**: Propagates the last or next valid observation, useful for data with temporal continuity.

- **Linear Interpolation**: Fills missing values using a linear function between known points.

- **K-Nearest Neighbors (KNN)**: Imputes missing values based on the average of the $k$ nearest neighbors.

- **MICE**: Multivariate Imputation by Chained Equations, an advanced statistical method for complex patterns.

- **SAITS**: Self-Attention based Imputation for Time Series, a state-of-the-art deep learning approach.

- **TimesNet**: Captures intra-period and inter-period variations with strong performance on imputation tasks.

## 2. ANOMALY HANDLING

- **Z-Score / IQR**: Statistical methods based on standard deviation or interquartile range to detect outliers.

- **Isolation Forest**: An efficient tree-based unsupervised algorithm for anomaly detection.

- **Local Outlier Factor (LOF)**: A density-based algorithm effective at detecting local anomalies.

- **One-Class SVM**: A semi-supervised method that learns a decision boundary around normal data points.

- **Autoencoder**: A neural network that reconstructs normal data; high reconstruction error indicates anomalies.

## 3. TRANSFORMATION

- **Log Transformation**: Stabilizes variance, often used for data with exponential growth.

- **Box-Cox Transformation**: A powerful transformation to correct for skewness and non-normality, requires positive data.

- **Yeo-Johnson Transformation**: An extension of Box-Cox that also supports zero and negative values.

- **Differencing**: First-order or seasonal differencing to remove trends or seasonality, making a series stationary.

## 4. DECOMPOSITION

- **Classical Decomposition**: Decomposes a series into trend, seasonal, and residual components using moving averages.

- **STL (LOESS)**: Seasonal-Trend decomposition using Loess, more robust and versatile than classical decomposition.

- **Fast Fourier Transform (FFT)**: Decomposes a signal into its constituent frequencies, ideal for strongly periodic series.

- **Wavelet Transform**: Provides a time-frequency representation, well-suited for non-stationary signals.

- **EMD / CEEMDAN**: Empirical Mode Decomposition and its variants, data-driven methods for nonlinear, non-stationary signals.

## 5. NORMALIZATION

- **StandardScaler**: Standardizes features to zero mean and unit variance, the default choice for many models.

- **MinMaxScaler**: Scales features to a specified range, typically $[0, 1]$.

- **MaxAbsScaler**: Scales each feature by its maximum absolute value, useful for sparse data.

- **RobustScaler**: Scales features using statistics robust to outliers (e.g., quantiles).

## 6. FORECASTING

- **ARIMA / SARIMA**: Classical statistical models for stationary series.

- **ETS (Exponential Smoothing)**: A family of forecasting methods based on weighted averages of past observations.

- **LightGBM / XGBoost**: Gradient boosting models, powerful when combined with feature engineering (e.g., lag features).

- **LSTM / GRU**: Recurrent Neural Networks for sequential data and long-term dependencies.

- **TCN (Temporal Convolutional Network)**: Uses convolutions for sequence modeling, often faster than RNNs.

- **Informer / Autoformer**: Efficient Transformer-based models for long-sequence forecasting.

- **PatchTST / iTransformer**: State-of-the-art Transformer models using channel-independent patching or inverted representations.

- **N-BEATS / N-HiTS**: MLP-based models with deep stacks of blocks, strong for univariate forecasting.

- **DLinear / TiDE**: Simple but powerful linear and MLP-based baselines.

- **TimesNet / TimeMixer**: Advanced state-of-the-art time-series models designed to capture diverse temporal patterns.

### A.3. Pseudocode

A.3.1. EXPLORATORY CONSTRUCTION OF FORECASTING STRATEGIES

The pseudocode of our Contrast-aware UCB Sampling for Exploratory Construction is shown as follows:

---

**Algorithm 1** Contrast-aware UCB Sampling

---

1: **Input:** Candidate toolchains $\mathcal{A}$; dataset meta-features $x$; empirical stats $\mu(a), \sigma(a)$ for $a \in \mathcal{A}$; coefficients $\lambda > 0$, $\varepsilon \in [0, 0.1]$; early-noise rounds $T_{\text{noise}}$; batch sizes $K_{\text{pos}}, K_{\text{neg}}$; total rounds $T_{\text{rounds}}$.
2: **Output:** Updated $\mu, \sigma$; Sampled set $\mathcal{D}$; logs.
3: Initialize $\mathcal{D} \leftarrow \emptyset$, Logs $\leftarrow \emptyset$
4: **for** $t = 1$ **to** $T_{\text{rounds}}$ **do**
5:     *// (1) UCB scoring*
6:     **for** each $a \in \mathcal{A}$ **do**
7:         $S(a) \leftarrow \mu(a) + \lambda \cdot \sigma(a)$
8:     **end for**
9:     *// (2) Early diversity noise*
10:     **if** $t \leq T_{\text{noise}}$ **then**
11:         **for** each $a \in \mathcal{A}$ **do**
12:             $S(a) \leftarrow S(a) + \text{Uniform}(-\varepsilon, +\varepsilon)$
13:         **end for**
14:     **end if**
15:     *// (3) Contrast-aware selection*
16:     $P \leftarrow \text{TopK by } S(a)$
17:     $U \leftarrow \text{TopK by } \sigma(a)$
18:     $L \leftarrow \text{BottomK by } S(a)$
19:     *// selects $K_{neg}$ negative candidates from union of low-score (L) and high-uncertainty (U) sets*
20:     $N \leftarrow \text{DiverseMerge}(U, L, K_{\text{neg}})$
21:     $\mathcal{S}_{\text{batch}} \leftarrow P \cup N$
22:     *// (4) Evaluate and update*
23:     **for** each $a \in \mathcal{S}_{\text{batch}}$ **do**
24:         $y \leftarrow \text{Evaluate}(a, x)$
25:         $\text{UpdateMeanStd}(\mu(a), \sigma(a); y)$
26:         $\mathcal{D} \leftarrow \mathcal{D} \cup \{(x, a, y)\}$
27:         Append log $(t, x, a, S(a), \mu(a), \sigma(a), y)$
28:     **end for**
29: **end for**

---

A.3.2. EXTRACTING STRATEGY LESSONS VIA CONTRASTIVE CAUSAL REASONING.

The pseudocode of our Extracting Strategy Lessons via Contrastive Causal Reasoning is shown as follows:

---

**Algorithm 2** Extracting Strategy Lessons

---

1: **Input:** Sampled set $\mathcal{D} = \{(x_i, a_i, y_i)\}_{i=1}^N$; threshold $\tau$; reasoning agent
2: **Output:** Structure Lesson library $\mathcal{L} = \{\phi_1, \dots, \phi_K\}$
3: *// (1) Split $\mathcal{D}$ into positive and negative outcomes*
4: $\mathcal{D}^+ \leftarrow \{(x_i, a_i) \mid y_i \geq \tau\}$
5: $\mathcal{D}^- \leftarrow \{(x_j, a_j) \mid y_j < \tau\}$
6: *// (2) Create contrastive tuples for each strategy*
7: **for** each strategy $a$ **do**
8: $\quad \mathcal{C}_a \leftarrow \{(x_i, x_j) \mid (x_i, a) \in \mathcal{D}^+, (x_j, a) \in \mathcal{D}^-\}$
9: **end for**
10: *// (3) Invoke LLM agent to extract lessons*
11: **for** each $\mathcal{C}_a$ **do**
12: $\quad$ Provide $\mathcal{C}_a$ to agent
13: $\quad$ Agent compares positive and negative meta-features under $a$
14: $\quad$ Agent generates candidate explanatory statements $\{\phi\}$ describing the conditions for success/failure
15: $\quad \mathcal{L} \leftarrow \mathcal{L} \cup \{\phi\}$
16: **end for**

---

A.3.3. LESSON VERIFICATION VIA CONTROLLED POLICY INTERVENTIONS.

The following pseudocode of our Lesson Verification explicitly implements controlled paired comparisons, ensuring that the only manipulated factor is the presence or absence of the lesson $\phi_k$, thus eliminating confounding effects during verification.

---

**Algorithm 3** Lesson Verification

---

1: **Input:** Lesson library $\mathcal{L} = \{\phi_1, \dots, \phi_K\}$; validation datasets with meta-features $\{x\}$; predefined success criterion $t$ (top-$\alpha\%$); confidence margin $\delta$
2: **Output:** Verified lesson set $\mathcal{L}^* \subseteq \mathcal{L}$
3: Initialize $\mathcal{L}^* \leftarrow \emptyset$
4: **for** each lesson $\phi_k \in \mathcal{L}$ **do**
5: $\quad$ *// Determine the validation datasets that satisfy the lesson condition $f_k$*
6: $\quad \mathcal{X}_k \leftarrow \{x \mid f_k(x) = \text{true}\}$
7: $\quad$ *// Decide whether $\phi_k$ is a positive (guiding) or negative (warning) lesson*
8: $\quad$ **if** $\phi_k$ is a positive (guiding) lesson **then**
9: $\quad\quad$ *// Controlled paired planning **with** lesson $\phi_k$*
10: $\quad\quad \mathcal{Y}_{\text{with}} \leftarrow \emptyset$
11: $\quad\quad$ **for** each $x \in \mathcal{X}_k$ **do**
12: $\quad\quad\quad$ Agent plans strategy guided by $\mathcal{L}$ (containing $\phi_k$)
13: $\quad\quad\quad$ Run forecasting with chosen strategy
14: $\quad\quad\quad y \leftarrow 1$ if result meets $t$; else $0$
15: $\quad\quad\quad$ Append $y$ to $\mathcal{Y}_{\text{with}}$
16: $\quad\quad$ **end for**
17: $\quad\quad P(y = 1 \mid f_k, \phi_k) \leftarrow \text{mean}(\mathcal{Y}_{\text{with}})$
18: $\quad\quad$ *// Controlled paired planning **without** lesson $\phi_k$ (all other conditions fixed)*
19: $\quad\quad \mathcal{Y}_{\text{without}} \leftarrow \emptyset$
20: $\quad\quad$ **for** each $x \in \mathcal{X}_k$ **do**
21: $\quad\quad\quad$ Agent plans strategy guided by $\mathcal{L} \setminus \{\phi_k\}$
22: $\quad\quad\quad$ Run forecasting with chosen strategy
23: $\quad\quad\quad y \leftarrow 1$ if result meets $t$; else $0$
24: $\quad\quad\quad$ Append $y$ to $\mathcal{Y}_{\text{without}}$
25: $\quad\quad$ **end for**
26: $\quad\quad P(y = 1 \mid f_k, \neg\phi_k) \leftarrow \text{mean}(\mathcal{Y}_{\text{without}})$
27: $\quad\quad$ *// Compute marginal causal effect under single-factor perturbation*
28: $\quad\quad \Delta_{\phi_k} \leftarrow P(y = 1 \mid f_k, \phi_k) - P(y = 1 \mid f_k, \neg\phi_k)$
29: $\quad$ **else**
30: $\quad\quad$ *// Negative lesson: compare **follow** vs. **contradict** $\phi_k$*

31:     $\mathcal{Y}_{\text{follow}} \leftarrow \emptyset$
32:     **for** each $x \in \mathcal{X}_k$ **do**
33:         Agent plans strategy guided by $\mathcal{L}$ and constrained to satisfy $\phi_k$
34:         Run forecasting with chosen strategy
35:         $y \leftarrow 1$ if result meets $t$; else 0
36:         Append $y$ to $\mathcal{Y}_{\text{follow}}$
37:     **end for**
38:     $P(y = 1 \mid f_k, \phi_k) \leftarrow \text{mean}(\mathcal{Y}_{\text{follow}})$
39:     $\mathcal{Y}_{\text{contra}} \leftarrow \emptyset$
40:     **for** each $x \in \mathcal{X}_k$ **do**
41:         Agent plans strategy under explicit intervention $\text{Contradict}(\phi_k)$
42:         Run forecasting with chosen strategy
43:         $y \leftarrow 1$ if result meets $t$; else 0
44:         Append $y$ to $\mathcal{Y}_{\text{contra}}$
45:     **end for**
46:     $P(y = 1 \mid f_k, \text{Contradict}(\phi_k)) \leftarrow \text{mean}(\mathcal{Y}_{\text{contra}})$
47:     *// Compute marginal causal effect under contradiction-based intervention*
48:     $\Delta_{\phi_k} \leftarrow P(y = 1 \mid f_k, \phi_k) - P(y = 1 \mid f_k, \text{Contradict}(\phi_k))$
49:   **end if**
50:   *// Retain lesson if effect exceeds confidence margin*
51:   **if** $\Delta_{\phi_k} > \delta$ **then**
52:     $\mathcal{L}^* \leftarrow \mathcal{L}^* \cup \{\phi_k\}$
53:   **else**
54:     Mark $\phi_k$ as low-confidence
55:   **end if**
56: **end for**

### A.3.4. FORECASTING WITH LESSON-GUIDED PLANNING.

The pseudocode of our Lesson-Guided Planning Forecasting procedure is shown as follows:

---

**Algorithm 4** Forecasting with Lesson-Guided Planning

---

1: **Input:** New dataset meta-features $x_{\text{new}}$; lesson library $\mathcal{L} = \{\phi_1, \ldots, \phi_K\}$; available toolchains $\mathcal{A}$
2: **Output:** Selected forecasting strategy $a^*$; evaluation result
3: *// (1) Retrieve relevant lessons*
4: $\mathcal{L}_x \leftarrow \{\phi_k \in \mathcal{L} \mid f_k(x_{\text{new}}) = \text{true}\}$
5: *// (2) Build structured prompt*
6: Prompt $\leftarrow$ encode $(x_{\text{new}}, \mathcal{L}_x, \mathcal{A})$ into LLM input format
7: *// (3) Agent reasoning and strategy planning*
8: Agent $\leftarrow$ Prompt
9: Agent outputs candidate strategy $a^*$
10: *// (4) Forecasting execution*
11: Run forecasting using $a^*$ on dataset with meta-features $x_{\text{new}}$
12: Collect predicted outputs and calculate evaluation result

---

## A.4. Prompt

**Extracting Strategy Lessons.** We use the following prompt for extracting Contrastive Causal Strategy Lessons:

```
You are an expert in causal analysis for time series forecasting. Given
    contrastive pairs under the same toolchain strategy a:

- Positives meta features: {Positive_meta}
- Negatives meta features: {Negative_meta}

Analyze the experimental results to extract causal lessons that explain why
```

```
        certain forecasting strategies succeed or fail under specific dataset
        characteristics.

Focus on identifying:
1. Causal relationships between dataset features and strategy performance
2. Necessary and sufficient conditions for strategy success
3. Causal mechanisms that explain the relationships
4. Confidence levels based on evidence strength

Your output should be Json format:
{
  "toolchain": "{Toolchain_name}",
  "important_features": "{Key_meta_features}",
  "description": "{Explanation}",
  "confidence": "{0.0-1.0}"
}

where "description" should be following condition (When), Result (Strategy
    succeeds / fails) and Causal explanation.

Here is an example:
Input: Positives meta features: {seasonal_peak = 24h, missing_rate = 0.05, CV =
    0.3}. Negatives meta features: {seasonal_peak = weak, missing_rate = 0.25, CV
    = 0.35}

Output: {
  "toolchain": "knn_none_none_fft_standard_patchtst",
  "important_features": "Strong seasonal peak, missing_rate < 0.1",
  "description": "[Phenomenon]: This strategy succeeds when seasonal structure is
      clear (e.g., a 24h seasonal peak) and missingness is low, because FFT
      isolates periodicity and PatchTST captures residuals. When missing_rate >
      0.2, imputation noise dominates and performance collapses. [Analysis]: Under
       low missingness, KNN imputation and FFT preprocessing preserve periodic
      structure for PatchTST; weak seasonality or high missingness breaks this
      chain and lowers reward on the same strategy runs.",
  "confidence": 0.85
}
```

**Lesson Verification.**    We use the following prompt for LLM strategy selection during lesson verification.

```
You are an expert in time-series forecasting and tool planning.
Your task is to select the best forecasting strategy for the validation dataset
    based on causal lessons learned, so that we can verify whether these lessons
    can improve the forecasting performance.

You are given:
- Meta features of dataset: {Meta_features}
- Causal lessons: {Lessons_description}
- Executable tool library (pick one tool per stage): {Stage_tool_library}

Select the most appropriate strategy for this dataset based on lessons. Consider:
1. Which lessons best match the dataset's characteristics?
2. Which strategies have the strongest verified causal effects?
3. How do the meta-features align with the lesson conditions?
4. Pick one tool per stage from the lists above

Your output should be Json format:
{
  "strategy": "{Selected_toolchain}",
  "reason": "{Explanation}"
}

Here is an example:
Input:
```

```
- Meta features of dataset:
  {seasonal_peak=24h, missing_rate=0.06, volatility=medium, KPSS=non-stationary}
- Causal lessons:
  1. Lesson1: [Phenomenon] KNN belongs to the Imputation tool category. When the
       dataset has a low missing rate, such as missing_rate=0.06, KNN tends to
       support strong forecasting performance. [Analysis] KNN estimates missing
       values from similar neighboring samples. With low missingness, most local
       references remain valid, so imputation noise is limited.
  2. Lesson2: [Phenomenon] FFT belongs to the Decomposition tool category. When
       the dataset has a clear 24-hour seasonal peak, FFT tends to improve
       forecasting performance. [Analysis] FFT extracts dominant frequency
       components. A 24-hour seasonal peak indicates stable daily periodicity,
       allowing FFT to provide clearer seasonal signals for downstream forecasting.

Output:
{
  "strategy": "knn_none_none_fft_standard_patchtst",
  "reason": "[Phenomenon]: The dataset has missing_rate=0.06 and seasonal_peak=24h
      , matching the KNN-imputation and FFT-decomposition lessons. [Causal
      Analysis]: KNN can fill the limited missing values with low noise, while FFT
       can extract the dominant daily periodic component. Therefore,
      knn_none_none_fft_standard_patchtst is selected."
}
```

**Lesson-Guided Planning.**   We use the following prompt for LLM strategy selection in final Lesson-Guided Planning.

```
You are an expert in time series forecasting and tool planning.
Your task is to plan the best forecasting strategy for the new dataset by meta
    features and lessons learned. You should adaptively match the meta-features of
     the dataset with lessons learned. And select the most appropriate strategy
    based on these lessons learned.

You have the memory of causal lessons, each of which describes
the conditions for the success or failure of certain prediction strategies.

You are given:
- Meta features of the new dataset: {Meta_features}
- Causal lessons: {Lessons_description}
- Executable tool library (pick one tool per stage): {Stage_tool_library}

Where causal lessons describe the conditions for the success or failure of certain
     prediction strategies.

Select the most appropriate tool at each stage from Executable tool library for
    this dataset based on lessons. Consider:
1. Which lessons best match the dataset's characteristics?
2. Which strategies have the strongest verified causal effects?
3. How do the meta-features align with the lesson conditions?

Your output should be Json format:
{
  "strategy": "{Selected_toolchain}",
  "reason": "{[Phenomenon]: describe how the key tool inside the selected strategy
       performs under this dataset's meta-features. [Causal Analysis]: explain why
       these meta-features affect this tool's behavior, based on its algorithmic
      properties.}"
}

Here is an example:
Input:
- Meta features of dataset:
  {seasonal_peak=24h, missing_rate=0.06, volatility=medium, KPSS=non-stationary}
- Causal lessons:
```

```
 1. Lesson1: [Phenomenon] KNN belongs to the Imputation tool category. When the
        dataset has a low missing rate, such as missing_rate=0.06, KNN tends to
        support strong forecasting performance. [Analysis] KNN estimates missing
        values from similar neighboring samples. With low missingness, most local
        references remain valid, so imputation noise is limited.
 2. Lesson2: [Phenomenon] FFT belongs to the Decomposition tool category. When
        the dataset has a clear 24-hour seasonal peak, FFT tends to improve
        forecasting performance. [Analysis] FFT extracts dominant frequency
        components. A 24-hour seasonal peak indicates stable daily periodicity,
        allowing FFT to provide clearer seasonal signals for downstream forecasting.

Output:
{
    "strategy": "knn_none_none_fft_standard_patchtst",
    "reason": "[Phenomenon]: The dataset has missing_rate=0.06 and seasonal_peak=24h
        , matching the KNN-imputation and FFT-decomposition lessons. [Causal
        Analysis]: KNN can fill the limited missing values with low noise, while FFT
         can extract the dominant daily periodic component. Therefore,
        knn_none_none_fft_standard_patchtst is selected."
}
```

### A.5. Understanding the Roles of Meta-features and Toolchain Diversity

In this section, we conduct a set of controlled analyses to disentangle the respective roles of meta-features and toolchain diversity in BECRA. Our goal is not to optimize performance under specific configurations, but to empirically validate the mechanistic dependencies underlying BECRA's lesson induction and verification process. All experiments in this section follow the same contrast-aware exploration, lesson induction, and causal verification pipeline as described in Section 3.

For clarity, we report two quantities throughout this section: (i) *extracted lesson candidates*, referring to candidate lessons produced by the induction procedure before verification, and (ii) *verified lessons*, referring to the subset of candidates that pass the causal verification tests. In Figures 3a and 3b, all counts are shown on a normalized scale for visualization. In particular, extracted lesson candidates are normalized by their value under the most informative setting in each experiment.

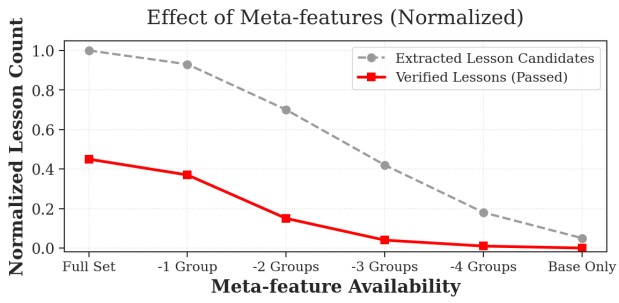
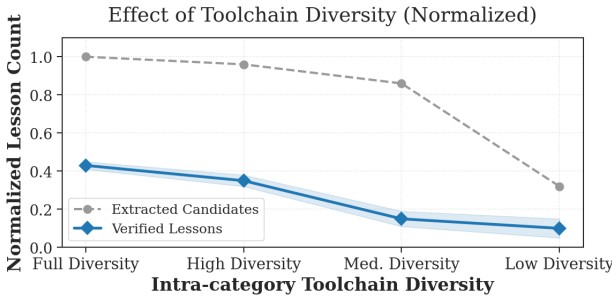

*(a)* **Effect of Meta-feature Availability.**        *(b)* **Effect of Toolchain Diversity.**

*Figure 3.* **Mechanistic dependencies underlying lesson induction in BECRA. (a) Effect of meta-feature availability** under a fixed and expressive toolchain space. As meta-features are progressively removed, the agent continues to extract candidate lessons from high-performing trajectories, but the number of lessons that pass causal verification collapses, indicating a failure to form conditionally valid explanations. **(b) Effect of intra-category toolchain diversity** under a fixed and informative meta-feature set. Reducing toolchain diversity leads to a gradual decline in both extracted candidates and verified lessons, reflecting a reduced availability of contrastive behaviors required for hypothesis testing rather than invalid explanations. Counts are normalized relative to the most informative setting in each experiment.

### A.5.1. FIXING TOOLCHAINS, VARYING META-FEATURE AVAILABILITY

We first study the role of meta-features by fixing the toolchain search space and progressively reducing the availability of meta-features used by the agent (Figure 3a).

**Experimental Setup**    We fix the toolchain library to the full set used in the main experiments, ensuring that the agent retains access to a rich and diverse space of compositional strategies, including preprocessing components, forecasting models, and post-processing operations.

We then progressively restrict the meta-feature set available to the agent. Rather than removing individual features arbitrarily, we perform group-wise ablation based on the semantic categories of dataset properties already used in BECRA, including: (i) missingness-related features, (ii) seasonality and periodicity features, (iii) trend and stationarity features, (iv) volatility and heteroscedasticity features, and (v) complexity and nonlinearity indicators. Concretely, we consider a sequence of settings from the full meta-feature set to progressively reduced sets ($-1/-2/-3/-4$ groups), as well as a *Base Only* setting that retains only the minimal meta-feature subset. At each stage, one or more feature groups are removed, reducing the agent's ability to characterize specific aspects of dataset structure while keeping the remaining meta-features unchanged.

**Observations**    As shown in Figure 3a, although contrast-aware exploration over the full toolchain space can still extract a non-trivial number of lesson candidates under reduced meta-feature availability, the number of verified lessons decreases sharply as meta-features are removed. In particular, when the meta-feature set becomes insufficient to describe key dataset properties, the agent is unable to associate observed performance differences with stable and conditionally valid explanations. As a result, many candidate lessons induced from high-performing trajectories fail the controlled intervention tests and are discarded during causal verification. This behavior reflects a breakdown in explanatory validity rather than a failure of exploration.

Importantly, this degradation is not necessarily reflected in short-term predictive performance during exploration. Instead, it manifests as a collapse in generalizable and verifiable lessons, indicating that meta-features primarily function as an interpretive scaffold that enables BECRA to abstract transferable knowledge, rather than as direct selectors of high-performing toolchains.

**Interpretation**    These results indicate that, even with a complete and expressive toolchain space, BECRA cannot reliably accumulate reusable lessons without sufficiently informative meta-features. Meta-features are therefore essential not for achieving isolated performance gains, but for supporting conditionally valid explanations and filtering out spurious correlations through causal verification.

### A.5.2. FIXING META-FEATURES, VARYING TOOLCHAIN DIVERSITY

We next examine the complementary role of toolchain diversity by fixing the meta-feature set and progressively restricting the toolchain search space (Figure 3b).

**Experimental Setup**    In this experiment, the full meta-feature set is preserved, allowing the agent to fully characterize dataset properties and apply the same lesson activation conditions as in the main method.

We then reduce the toolchain space while preserving functional coverage. Specifically, we subsample the toolchain library such that each major functional category (e.g., preprocessing, model family, post-processing) remains available, while intra-category diversity is reduced. This procedure limits the range of inductive biases instantiated by the toolchains while avoiding degenerate settings in which essential processing steps are entirely removed. We report results across four levels of toolchain diversity (Full / High / Medium / Low), following the x-axis in Figure 3b. All other components of BECRA, including contrast-aware exploration and causal verification, remain unchanged.

**Observations**    As toolchain diversity is progressively reduced, Figure 3b shows a gradual decline in both extracted lesson candidates and verified lessons. Compared to meta-feature ablation, the reduction in verified lessons is smoother under moderate diversity loss and becomes pronounced only in the low-diversity regime. With informative meta-features still available, the agent remains capable of reasoning about dataset properties and formulating candidate explanations. However, the reduced diversity of toolchains limits the range of observable contrasts between successful and unsuccessful trajectories, thereby constraining the agent's ability to test and refine alternative hypotheses about strategy effectiveness.

Consequently, fewer distinct lessons can be induced and verified, not because explanations become invalid, but because the available toolchain space provides insufficient behavioral variation to support richer causal comparisons.

**Interpretation** These findings suggest that toolchain diversity primarily governs the breadth of contrastive evidence available to the agent. While meta-features determine the conditions under which strategies may be effective, toolchain diversity controls how many such strategies can be meaningfully compared and validated through contrast-aware exploration.

A.5.3. DISCUSSION: COMPLEMENTARY ROLES OF META-FEATURES, TOOLCHAINS, AND LESSONS

Taken together, the analyses in Figures 3a and 3b clarify the distinct yet complementary roles of meta-features, toolchain diversity, and lessons in BECRA. Meta-features provide a structured view of dataset-level properties that enables the agent to contextualize observed outcomes and formulate conditionally valid explanations. Toolchain diversity instantiates heterogeneous inductive biases, creating the contrastive structure required for identifying why certain strategies succeed or fail under specific conditions. Lessons emerge as conditional abstractions that link these two components and are retained only when their causal effect is confirmed through controlled policy interventions.

Crucially, neither meta-features nor toolchains alone are sufficient to support transferable knowledge acquisition. Without informative meta-features, successful toolchains cannot be abstracted into reusable lessons; without sufficient toolchain diversity, meta-feature–conditioned hypotheses cannot be adequately tested. These results reinforce that BECRA's contribution lies not in any specific configuration of meta-features or toolchain components, but in a general agent training paradigm that integrates contrast-aware exploration, conditional lesson induction, and causal verification. By explicitly modeling dataset-level properties through meta-features and leveraging a compositional toolchain space to generate contrastive evidence, BECRA enables agents to acquire transferable and verifiable lessons, directly addressing the limitations of static or performance-driven pipeline selection highlighted in the Introduction.

### A.6. Computational Cost Analysis

To analyze the computational efficiency of our paradigm, we compare BECRA against a representative AutoML framework. To ensure a fair comparison, we standardized the set of candidate toolchains for both methods. The cost of a single trial—defined as a full train-evaluation cycle for one toolchain on a given prediction horizon—was held constant. Our primary metric for computational cost is therefore the total number of trials required to find a high-performing strategy. For the AutoML baseline, we employed a Bayesian Optimization framework, a powerful and widely-used strategy for efficient search. For each new dataset, the AutoML process was initiated from scratch and run until its performance on a validation set converged.

*Table 5.* Comparison of trial costs for AutoML (cold start) vs. BECRA (warm-started). BECRA(Contrast-aware UCB) needs a warm start on the dataset, meaning that the agent first explores on one (or a few) dataset to collect causal lessons and to initialize the sampling statistics $\mu(a)$ and $\sigma(a)$. In contrast, AutoML has to perform a cold start from scratch on every dataset. We totally have 6 datasets(ETTh1, ETTh2, ETTm1, ETTm2, Electricity, Weather). **Warm-start trials** denote the number of trials BECRA requires on a single dataset, summed over the four prediction horizons. For BECRA, **New dataset trials** denotes the sum of trials conducted on each of the remaining 5 datasets over four prediction lengths, averaged over the choice of warm-start dataset. For AutoML, it denotes the sum of trials for all 6 datasets under cold start. **Total trials** denotes total trials for all 6 datasets; for BECRA, the value is averaged over the choice of warm-start dataset.

| Method | Warm-start trials | New dataset trials | Total trials |
|---|---|---|---|
| AutoML (cold start) | — | 1177 | 1177 |
| BECRA (warm-started) | 158 | 53 | 211 |

Table 5 reports the comparative trial costs. The AutoML mechanism has no warm start, thus requiring a total of 1,177 trials on 6 datasets. In contrast, BECRA can have an initial warm start on a single dataset with 158 trials, after which $\mu(a)$, $\sigma(a)$ and lessons can transfer to new datasets. For the remaining 5 datasets, BECRA only requires 53 trials in total. Both BECRA trials in warm-start and the remaining datasets are averaged results, because warm-start can begin in any of the 6 datasets. Therefore, BECRA reduces total trials by 5.6× (82%) compared to AutoML (1177 vs 211).

This disparity stems from a fundamental difference in paradigm. AutoML performs a one-shot, task-specific search, meaning its high computational cost is incurred repeatedly for every new dataset. It lacks a mechanism for accumulating or transferring knowledge across tasks, making it inherently unscalable in dynamic environments. BECRA, on the other hand, operates on a principle of knowledge accumulation and amortization. It invests a one-time, offline computational budget during its exploration phase to build a reusable, symbolic knowledge base of "lessons". This initial investment is then amortized over a lifetime of subsequent tasks. For any new dataset, BECRA bypasses the expensive search process entirely,

instead leveraging its accumulated knowledge for near-instantaneous, low-cost planning. As the number of forecasting tasks grows, the marginal cost per task for BECRA approaches zero, an advantage that is particularly crucial in industrial settings with hundreds of evolving time-series.

Therefore, our analysis concludes that BECRA is not only superior in terms of predictive performance but also represents a far more scalable and economical paradigm than conventional AutoML systems. By shifting the focus from repetitive searching to reusable reasoning, BECRA is significantly better suited for the demands of large-scale and dynamic real-world forecasting environments.

### A.7. Uncertainty estimation and statistical reliability

Experiments have been run for multiple independent trials. Margin of Error (MOE) with 95% confidence intervals were included in the Table 6, ensuring that the performance differences can be evaluated for statistical reliability.

MOE quantifies uncertainty stemming from randomness in model initialization, data sampling, or stochastic training dynamics, and directly reflects the width of the corresponding confidence interval. A smaller MOE indicates that the sample mean is tightly concentrated around the true mean, while a larger MOE signals greater statistical uncertainty. This relationship allows differences between models to be interpreted in terms of statistical significance rather than raw fluctuations.

**Margin of Error (MOE) Calculation**

$$\text{MOE} = z_{\alpha/2} \times \left( \frac{\sigma}{\sqrt{n}} \right) \tag{1}$$

**Notation**

- $z_{\alpha/2}$: Critical value for the desired confidence level (e.g., 1.96 for 95% confidence)

- $\sigma$: Sample standard deviation across repeated runs

- $n$: Number of runs used to estimate uncertainty

*Table 6.* MOE comparison across models.

| Dataset | BECRA | | TimeMixer | | TimesNet | | PatchTST | | iTransformer | |
|---|---|---|---|---|---|---|---|---|---|---|
| | MOE (MSE) | MOE (MAE) | MOE (MSE) | MOE (MAE) | MOE (MSE) | MOE (MAE) | MOE (MSE) | MOE (MAE) | MOE (MSE) | MOE (MAE) |
| ETTH1 | 3.11E-03 | 2.09E-03 | 3.19E-03 | 2.18E-03 | 6.98E-03 | 4.59E-03 | 1.80E-03 | 9.73E-04 | 4.62E-04 | 6.03E-04 |
| ETTH2 | 4.26E-03 | 2.62E-03 | 3.25E-03 | 2.97E-03 | 4.37E-03 | 2.62E-03 | 7.58E-03 | 4.10E-03 | 1.83E-03 | 7.77E-04 |
| ETTm1 | 3.75E-03 | 2.40E-03 | 4.84E-03 | 4.56E-03 | 4.94E-03 | 2.61E-03 | 2.86E-03 | 1.53E-03 | 2.38E-03 | 8.97E-04 |
| ETTm2 | 1.66E-03 | 1.07E-03 | 1.73E-03 | 6.92E-04 | 2.88E-03 | 7.86E-04 | 1.27E-03 | 1.21E-03 | 7.78E-04 | 1.59E-03 |
| Electricity | 4.08E-04 | 4.50E-04 | 4.29E-04 | 6.01E-04 | 6.86E-04 | 6.78E-04 | 1.96E-04 | 3.28E-04 | 3.21E-04 | 1.93E-04 |
| Weather | 1.10E-03 | 9.79E-04 | 9.05E-04 | 6.13E-04 | 2.34E-03 | 1.64E-03 | 3.76E-04 | 8.04E-04 | 7.67E-04 | 8.63E-04 |

We report multi-run uncertainty (95% CI / MOE) for methods whose outcomes are affected by stochastic training (e.g., BECRA and trainable baselines such as TimeMixer, TimesNet, PatchTST, and iTransformer), using $n = 5$ independent random seeds. For fixed-checkpoint foundation models evaluated with deterministic inference (e.g., Chronos, Moirai/Moirai-2.0, Time-LLM, and GPT4TS; with dropout disabled and decoding fixed), repeated runs yield (near-)identical predictions, making across-run confidence intervals degenerate and uninformative. For closed-source or hosted LLM APIs (e.g., ChatGPT-4 and Gemini, and LLaMA models accessed via hosted APIs), repeated calls may conflate serving-side nondeterminism and model-version drift that are not controllable (e.g., no exposed random seed), thus the resulting variance is not directly comparable to training-seed variance; we therefore use deterministic decoding and report a single run for these models.

### A.8. Scope and Discussion

**BECRA and Time-Series Foundation Models: Applicability and Paradigm-Level Limitations**

BECRA is designed to study agent-level strategy induction and causal verification over heterogeneous, compositional toolchains, rather than to analyze or compare the internal behavior of monolithic forecasting models. A core assumption underlying BECRA is that strategy effectiveness can be meaningfully explained through observable contrasts between alternative toolchain configurations and conditionally activated by dataset-level meta-features. This assumption requires strategy choices to be modular, comparable, and amenable to controlled intervention.

In contrast, time-series foundation models (TSFMs) are typically deployed as pre-trained, end-to-end black-box models whose performance differences primarily stem from factors such as model scale, pretraining data, and training objectives, rather than from explicitly decomposable or interpretable strategy components. Due to their opaque internal mechanisms and limited controllability, it is generally not possible to isolate which aspects of a TSFM's behavior are responsible for its performance under specific dataset conditions, nor to associate such behavior with meta-feature–conditioned explanations in a causally testable manner.

As a result, BECRA is not well-suited for inducing lessons that aim to explain why different TSFMs perform differently, since such performance differences cannot be reliably attributed to conditionally valid strategies that can be verified through intervention-style comparisons. This limitation does not reflect a weakness of BECRA, but instead follows directly from the paradigm-level assumptions required for lesson-based agent training.

Accordingly, BECRA is intended for settings in which strategy choices are themselves interpretable, compositional, and amenable to causal analysis—conditions that are necessary to support contrast-aware exploration, lesson induction, and causal verification. Time-series foundation models, by contrast, are typically used as holistic predictive systems whose behavior is not naturally expressed at the level of decomposable strategies or intervention-ready decision units, and therefore do not satisfy the assumptions required for lesson-based strategy induction.

**Potential lesson conflicts under segmented meta-features.** BECRA relies on causal lessons that map dataset meta-features to recommended toolchain components. In this work, meta-features are computed once at the dataset level, yielding a single global characterization. Under this assumption, when meta-features are internally consistent, the retrieved lessons typically align without conflict. A natural extension is to compute *segmented* meta-features for long or non-stationary series, where different temporal segments may exhibit different characteristics. In such cases, lessons activated by different segments may provide inconsistent guidance, making explicit conflict handling and aggregation strategies for retrieved lessons an important direction for future work. Such cross-segment consolidation echoes hierarchical local-to-global aggregation in long-context modeling (Zeng et al., 2026).

**Human-in-the-loop extension for high-stakes deployment.** While BECRA includes an automatic Lesson Verification stage to filter unreliable or spurious lessons, in high-stakes settings (e.g., safety-critical or high-cost decision making) it can be beneficial to introduce an optional human review step before committing lessons to the knowledge base. This additional safeguard mitigates rare failure modes such as hallucinated rationales or brittle causal associations, at the cost of reduced automation and increased latency. Exploring systematic criteria and workflows for such human-in-the-loop vetting (e.g., reviewing only borderline-confidence lessons) is a pragmatic direction for improving reliability while managing the efficiency–accuracy trade-off.

### A.9. Experiments Setup Details

**Device.**    Dual NVIDIA GeForce RTX 5090 GPUs.

**Datasets and Tasks.**    Seven public benchmarks are used: ETTh1, ETTh2, ETTm1, ETTm2, Electricity, Weather, and M4. M4 is used for short-term forecasting, while the other datasets are used for long-term forecasting.

- **ETTh1**
    - Domain: Electricity transformer temperature and load.
    - Size: ∼17,420 time steps.
    - Sampling frequency: Hourly.
    - Feature columns: 6 additional variables (after removing time and target).

- **ETTh2**
    - Domain: Electricity transformer temperature and load.

- Size: ~17,420 time steps.
- Sampling frequency: Hourly.
- Feature columns: 6 additional variables.

- **ETTm1**

  - Domain: Electricity transformer temperature and load.
  - Size: ~69,680 time steps.
  - Sampling frequency: 15 minutes.
  - Feature columns: 6 additional variables.

- **ETTm2**

  - Domain: Electricity transformer temperature and load.
  - Size: ~69,680 time steps.
  - Sampling frequency: 15 minutes.
  - Feature columns: 6 additional variables.

- **Electricity**

  - Domain: Electricity consumption across multiple clients.
  - Size: 26,304 time steps $\times$ 321 series.
  - Sampling frequency: Hourly.
  - Feature columns: 320 additional series (each treated as a feature in multivariate forecasting).

- **Weather**

  - Domain: Meteorological measurements from the Max Planck Institute.
  - Size: 52,696 time steps.
  - Sampling frequency: 10 minutes.
  - Feature columns: 20 additional meteorological variables.

- **M4**

  - Domain: Mixed real-world economic, demographic, financial, and industrial time series.
  - Size: 100,000 univariate series.
  - Sampling frequency: Mixed (yearly, quarterly, monthly, weekly, daily, hourly).
  - Feature columns: 0 (strictly univariate).

All models operate on consistent data splits following standard practice for each dataset.

**Baselines Used.** Below is a list of baseline models used in the experiment.

1. **ChatGPT-4** (Achiam et al., 2023): Used as an untrained forecasting agent without applying the BECRA paradigm; it directly plans the toolchain based on dataset meta-features and available tools. This baseline separates intrinsic LLM capabilities from BECRA's structured reasoning improvements.

2. **Gemini** (Team et al., 2023): Same usage and purpose as ChatGPT-4.

3. **LLaMA** (Touvron et al., 2023): Same usage and purpose as ChatGPT-4.

4. **TimeMixer** (Wang et al., 2024a): A decomposable multi-scale mixing architecture capturing temporal patterns at different frequencies.

5. **TimesNet** (Wu et al., 2022): A temporal 2D-variation modeling framework converting time series into multi-scale 2D representations.

6. **PatchTST** (Nie et al., 2022): A patch-based Transformer applying channel-independent self-attention over time-series patches.

7. **iTransformer** (Liu et al., 2023): An inverted Transformer modeling series along the feature dimension instead of the temporal dimension.

8. **Time-LLM** (Jin et al., 2023): A reprogramming-based approach adapting large language models for time-series forecasting.

9. **GPT4TS** (Zhou et al., 2023): A generative LLM-based forecaster leveraging pretrained language models to predict future values.

10. **Chronos** (Ansari et al., 2024): A pretrained probabilistic time-series foundation model trained on large-scale datasets.

11. **Moirai** (Woo et al., 2024): A modular instruction-routing architecture with strong generalization across diverse tasks including time-series reasoning.

**Hyper-Parameters.** For long-term forecasting, the input sequence length is 96, with prediction horizons of 96, 192, 336, and 720. For M4, the input lengths are 12 and 96, and the prediction lengths are 6 and 48.

This configuration follows the dominant evaluation protocol in recent forecasting literature. Mainstream models such as TimesNet (Wu et al., 2022), iTransformer (Liu et al., 2023), and TimeMixer (Wang et al., 2024a) adopt the same settings, ensuring comparability and consistency with prior studies.

**Evaluation Protocol.** All long-term forecasting results are evaluated using Mean Squared Error (MSE) and Mean Absolute Error (MAE):

$$\text{MSE} = \frac{1}{N} \sum_{i=1}^{N} (y_i - \hat{y}_i)^2$$

$$\text{MAE} = \frac{1}{N} \sum_{i=1}^{N} |y_i - \hat{y}_i|$$

**Notation**

- $N$: Number of forecasted data points
- $y_i$: Ground-truth value at index $i$
- $\hat{y}_i$: Predicted value at index $i$

For short-term forecasting (M4), the evaluation uses SMAPE, MASE, and OWA, the official M4 metrics:

$$\text{SMAPE} = \frac{100\%}{N} \sum_{i=1}^{N} \frac{|y_i - \hat{y}_i|}{(|y_i| + |\hat{y}_i|)/2}$$

$$\text{MASE} = \frac{\frac{1}{N} \sum_{i=1}^{N} |y_i - \hat{y}_i|}{\frac{1}{N-m} \sum_{j=m+1}^{N} |y_j - y_{j-m}|}$$

$$\text{OWA} = \frac{1}{2} \left( \frac{\text{SMAPE}}{\text{SMAPE}_{\text{naive2}}} \right) + \frac{1}{2} \left( \frac{\text{MASE}}{\text{MASE}_{\text{naive2}}} \right)$$

**Notation**

- $N$: Number of forecasted points

- $m$: Seasonal period

- $y_i$: Ground-truth value

- $\hat{y}_i$: Predicted value

- $\text{SMAPE}_{\text{naive2}}$: SMAPE of the seasonal naïve baseline

- $\text{MASE}_{\text{naive2}}$: MASE of the seasonal naïve baseline

