# OpenReview forum: "Bootstrapped Exploration with Causal Reasoning: A Training Paradigm for Adaptive Forecasting Agent"
_ICML.cc/2026/Conference — ICML 2026 regular_

### Official Review · Reviewer_i3BV · 2026-03-05

**Soundness:** 3
**Presentation:** 3
**Significance:** 3
**Originality:** 3
**Overall Recommendation:** 4
**Confidence:** 3

**Summary:**

The paper introduces BECRA, an unsupervised training paradigm designed to create adaptive time-series forecasting agents. Unlike traditional methods that rely on static, manually tuned frameworks or black-box foundation models, BECRA enables agents to autonomously learn and accumulate transferable strategy knowledge. The framework operates through a four-stage cycle: contrast-aware exploration (using UCB sampling to find both successes and failures), causal lesson extraction (distilling symbolic rules via LLM-based induction), lesson verification (using controlled policy interventions), and lesson-guided planning for zero-shot adaptation to unseen datasets. Experimental results demonstrate that BECRA outperforms specialized single-architecture models and untrained LLM agents while significantly reducing the computational costs associated with per-task searches.

**Compliance With Llm Reviewing Policy:**

Affirmed.

**Final Justification:**

The paper proposes BECRA, a novel training paradigm for adaptive time-series forecasting agents that accumulate reusable strategy-level knowledge. The idea original and practically relevant, particularly in addressing the limitations of static pipelines and per-dataset search. The approach is generally well-motivated and supported by empirical results, although some aspects (e.g., reliance on handcrafted meta-features and exploration cost) remain limitations.

My main concern in the initial review was the overstatement of causal claims; the rebuttal clarified that causality is defined at the level of agent decision-making via controlled interventions, which resolves this issue. The additional discussion on meta-features, failure cases, and qualitative lessons also improves clarity and confidence in the method.

I will give a weak accept recommendation.

**Key Questions For Authors:**

1. How sensitive is the lesson extraction process to the specific performance threshold ($\tau$) used to partition successful and unsuccessful outcomes?
2. Since the agent relies on dataset meta-features to retrieve relevant lessons, how sensitive is the framework to the choice or quality of these meta-features?
3. Can the authors provide qualitative examples of learned strategy lessons and analyze whether they align with known forecasting principles?
4. How does BECRA handle "conflicting" lessons that might be generated from different datasets that share similar meta-features but require different strategies?

**Limitations:**

1. The framework still relies on several heuristic design choices, including handcrafted meta-features, manually curated toolchains, and exploration strategies, which somewhat limits the level of generalizability.
2. There is a potential for "hallucinated" causal links if the LLM reasoner identifies spurious correlations between meta-features and toolchain performance during the induction phase.
3. The exploration stage requires evaluating multiple pipelines, which may lead to non-trivial computational cost during training.

**Strengths And Weaknesses:**

**Strengths**

1. The paper addresses an important practical problem in time-series forecasting: selecting appropriate forecasting pipelines across heterogeneous datasets without costly manual tuning.
2. The proposed BECRA paradigm combines several ideas (agent exploration, contrastive reasoning, symbolic lesson extraction, and verification) into a coherent framework for accumulating reusable forecasting strategies.
3. The experimental evaluation spans multiple benchmark datasets and compares against several categories of baselines (LLM agents, deep forecasting models, and foundation models), suggesting reasonable empirical performance.

**Weaknesses**
1. The claim of causal reasoning appears overstated. The method measures policy-level performance differences when lessons are enforced, but this does not establish causal relationships in the underlying data-generating process.
2. The meta-feature representation is relatively simplistic, relying on hand-crafted scalar statistics that may not capture complex temporal dynamics, regime shifts, or high-dimensional dependencies in time-series data.
3. The paper lacks analysis of failure cases, which would help understand when the agent’s strategy selection performs poorly or when lessons transfer incorrectly.

---

> ### Author Rebuttal · Authors · 2026-03-31
>
> We sincerely thank reviewer for the constructive comments. Below we address each concern that we will make more explicit in the revision. We also provide figures and tables [here](https://shorturl.at/KqyV6).
>
> # Weakness 1: Causal Reasoning
> We understand this point requires careful wording. In BECRA, the causal claim is not about identifying the underlying causal structure of the data-generating process. Rather, it is defined at the agent decision-making level. In Section 3.3, our verification uses paired rollouts on the same dataset with identical tool sets and decoding settings, differing only in whether lesson φ_k is injected. The effect Δ_{φ_k}=P(y=1|f_k,φ_k)−P(y=1|f_k,¬φ_k) then measures the intervention effect of that lesson on the agent's decision outcome. Lessons will be retained only when Δ is consistently positive.
>
> In BECRA, this verification process confirms the meta-feature→strategy→outcome mapping as a reusable principle, following the interventionist causation [1]. We will revise the paper to clarify this scope.
>
> # Weakness/Question 2: Meta-Feature
>
> We understand the concern for meta-features. We would like to clarify that
> - First, the meta-features in our experiments follow **established practice [2]**.
> - Second, the meta-features include **change-point features via PELT** for regime shifts (Appendix A.1.6), **spectral/Fourier peak features** for periodic structure (Appendix A.1.5), **ACF/PACF-based features** for temporal dependence (Appendix A.1.5), and **entropy-based features** for noise/complexity (Appendix A.1.9). They captures most factors raised by reviewer.
>
> Our claim is not that these meta-features fully characterize all time-series dynamics, but that they provide a practical and interpretable signal. Appendix A.5 further shows that removing meta-feature groups sharply reduces the number of verified lessons, supporting their functional importance in BECRA.
>
> # Weakness 3 & Questions 3-4
>
> ## Link for Fig.4, Fig.5 and Table7: https://anonymous.4open.science/r/i3BV/1.pdf
>
> > Failure cases are shown in Fig.4. One filtered case linked PatchTST to series length>10000, but it was spurious since series also differed in volatility and seasonality, leading to no consistent improvement (Δ_{φ_k}<δ). Another claimed STL decomposition is always beneficial, which degraded performance on datasets without clear periodicity. These cases show that the verification stage filters plausible but spurious associations. We will add such failure-case analysis in the revision.
>
> > Qualitative lessons are provided in Fig.5. In these examples, BECRA learns that TimesNet is more suitable for block-wise missingness since it requires training, whereas Linear Interpolation performs better for point-wise missingness since it is training-free. They complement each other, suggesting induced knowledge is meaningful for adaptive strategy selection, rather than arbitrary.
>
> > Conflicting lessons may arise during knowledge construction even if different datasets with similar meta-features support divergent strategies. Our mechanism for handling this is **Verification**. Each φ_k is evaluated via paired rollouts across datasets satisfying its condition; inconsistent effects attenuate Δ_{φ_k} below δ, leading to removal (Table 3, Verification Ablation). We are not suggesting conflicts are impossible, but rather BECRA is designed to reduce the retention of contradictory or unstable lessons. We will clarify this more explicitly in the revision.
>
> # Question 1
> Detailed sensitivity summary is provided in Table7. Empirically, τ governs an efficiency-coverage trade-off, not lesson quality. When it is too low (0.1-0.2), fewer contrastive pairs yield fewer verified lessons. When τ is too high (0.4-0.5), more candidates enter induction, but the final number of verified lessons remains comparable to default 0.3. It is because verification filters low-quality candidates. In our experiments, τ=0.3 provides the best balance, yielding the highest number of verified lessons with optimal cost.
>
> # Limitations
> We understand the concern of design choices. As noted in Weakness reply, the proposed paradigm BECRA provides a mechanism for accumulating, validating, and reusing strategy knowledge rather than re-solving each dataset independently.
>
> For hallucinated lessons, it is exactly what our verification (Section 3.3) addresses. Ablation study in Table 3 empirically confirms its effectiveness in filtering these lessons.
>
> Finally, regarding training-time cost, we note that BECRA reduces repeated cold-start search by amortizing strategy acquisition across tasks. Appendix A.6 shows a 5.6× reduction in trials (82%) relative to cold-start AutoML, recognized by Reviewers GHud and XGwo. We agree that training BECRA has non-trivial upfront cost, and we will make this trade-off clearer in the revision.
>
> [1] Woodward, J. (2005). Making things happen: A theory of causal explanation.
>
> [2] Talagala, T. S. (2023). Meta-learning how to forecast time series.

---

> > ### Author Rebuttal · Reviewer_i3BV · 2026-04-03
> >
> > Thank you for the detailed rebuttal. I appreciate the clarifications and additional analysis, which address several of my earlier concerns.
> >
> > In particular, the revised explanation of “causality” as agent-level interventional effects (rather than claims about the data-generating process) is much clearer and resolves my main concern about overstatement. The added discussion on meta-features and their practical role is also helpful, and the qualitative examples and failure cases provide useful insight into how the framework behaves.
> >
> > While some limitations remain (e.g., reliance on handcrafted meta-features and exploration cost), I am satisfied that the paper now presents a more clearly scoped and well-supported contribution. I will update my recommendation to weak accept.

---

### Official Review · Reviewer_6y98 · 2026-03-10

**Soundness:** 4
**Presentation:** 3
**Significance:** 3
**Originality:** 3
**Overall Recommendation:** 4
**Confidence:** 3

**Summary:**

This paper proposes a novel agent training paradigm for time-series forecasting that learns reusable strategy knowledge via exploration, contrastive causal reasoning, and lesson verification, enabling zero-shot adaptation across datasets.
The effectiveness of BECRA is demonstrated on multiple time-series benchmarks, showing improvements over several baseline forecasting models, LLM-based agents, and time-series foundation models.
Additional ablation studies and robustness experiments further analyze the contributions of the proposed components.

**Compliance With Llm Reviewing Policy:**

Affirmed.

**Final Justification:**

The rebuttal has addressed my concerns. Adding the efficiency analysis would increase the feasibility, so I have maintained my positive score and increased the 'soundness' score to 4.

**Key Questions For Authors:**

Please refer to **Weaknesses**.

**Limitations:**

Yes

**Strengths And Weaknesses:**

**Strengths:**
- It reframes time-series forecasting pipeline construction as an agent training problem. This provides an interesting perspective by shifting from optimizing fixed pipelines to learning an adaptable system that combines preprocessing and forecasting tools.
- The use of symbolic 'causal lessons' may improve transparency compared to purely model-driven approaches.
- The contrast-aware UCB sampling strategy provides a principled way for collecting both successful and failed strategies. This design enables meaningful contrasts that support strategy induction.
- The ablation studies and experiments are comprehensive.

**Weaknesses:**
- The efficiency analysis compares the number of trials against AutoML, but it does not quantify the costs, API latency, or scalability implications incurred by the LLM during the extensive reasoning and verification phases.
- Although the paper frequently refers to 'causal reasoning', the proposed method mainly relies on performance contrasts and controlled policy interventions. As clarified in Section 3.3, the causality is at the 'agent decision-making' level. It largely depends on the LLM's pattern matching rather than strict statistical causal inference.
- The Reference Section should be restructured for consistency. Several ArXiv papers are cited using inconsistent formats, such as paper 'Timecopilot' and 'From News to Forecast: Integrating Event Analysis in LLM-based Time Series Forecasting with Reflection'.

---

> ### Author Rebuttal · Authors · 2026-03-31
>
> We sincerely thank reviewer 6y98 for the constructive comments and positive assessment of the paper’s novelty, technical soundness, and experimental comprehensiveness. Below we address the three concerns.
>
> ## Weakness 1: Efficiency analysis does not quantify LLM costs, API latency, or scalability
>
> We thank the reviewer for this suggestion and agree that the efficiency discussion should be more complete.
>
> Our tool library (Appendix A.2) covers 6 pipeline stages and includes 35 tools in total. For warm-start exploration plus transferring to new datasets, the LLM API cost of BECRA is approximately $28–35 under GPT-4 pricing. Additionally, as shown in our transferability analysis (Table 4), the lesson memory can be leveraged by open-source LLMs such as LLaMA-3.1-70B and Qwen-2.5-72B with only marginal performance degradation, which means that API costs can be in principle eliminated via local deployment.
>
> It is worth noting that although AutoML does not incur any API costs, it requires 5.6× more toolchain trials than BECRA (1177 vs. 211, Table 5). Each trial consumes a significant amount of GPU computing resources. These trials represent a huge hidden monetary cost, especially in industrial scenarios where hundreds of evolving time series pipelines need to be maintained. BECRA instead incurs a low reasoning cost to acquire transferable strategy knowledge, which reduces repeated search on future datasets.
>
> Regarding latency and scalability, each LLM call includes a brief prompt and compact structured output, while the dominant runtime comes from executing candidate toolchains themselves within a few seconds. Therefore, compared to the GPU time in each trial (the toolchain execution), the total inference time of LLM API is negligible. In addition, the LLM usage also scales favorably as the number of datasets grows. Lesson extraction and verification are one-time costs during warm-start. After that, the learned lessons and sampling statistics $\mu(a)$, $\sigma(a)$ can be directly transferred to new datasets, and each dataset only requires a few planning calls. As a result, the per-task LLM overhead is effectively amortized and it will not increase with the number of downstream datasets.
>
> **We will add these discussions to the revision to make a more comprehensive efficiency analysis.**
>
> ## Weakness 2: The use of "causal reasoning" terminology
>
> We appreciate this important comment and agree with the reviewer’s interpretation. BECRA does not perform formal statistical causal inference on the underlying time-series data-generating process. Rather, as noted in Section 3.3, the causal interpretation in our paper is at the agent decision-making level. Specifically, our verification stage (Algorithm 3) instantiates Woodward's interventionist account of causation [1]. Each lesson $\phi_k$ is evaluated through controlled interventions on the agent's decision policy: all execution conditions remain fixed, only the presence or absence of $\phi_k$ is manipulated. This is to satisfy Woodward's definition of an ideal intervention.
>
> A lesson is retained only if $\Delta_{\phi_k} > \delta$ across paired rollouts, which corresponds to the invariance requirement that a genuine causal relationship should remain stable under repeated interventions. Additionally, we verify negative lessons by forcing the agent to contradict them, thereby exploring counterfactual dependence in both directions, which goes beyond standard pattern matching.
>
> For LLM pattern matching, LLM is used in the lesson induction stage (§3.2) to generate candidate hypotheses from contrastive successes and failures. These candidate lessons are then subjected to empirical verification: under matched execution conditions, we compare agent behavior and downstream performance with and without injecting a lesson. Thus, a lesson is retained not because the LLM identifies a plausible pattern, but because enforcing that lesson yields a consistent positive intervention effect in the agent’s planning process.
>
> We agree that the phrase "causal reasoning" may suggest a stronger claim than what is intended. In the revision, we will clarify this terminology more explicitly in the introduction and method section, emphasizing that our use of "causal" refers to controlled intervention on agent policy decisions, rather than causal discovery over the underlying data-generating mechanism.
>
> [1] Woodward, J. (2005). Making things happen: A theory of causal explanation. Oxford university press.
>
> ## Weakness 3: Reference formatting inconsistency
> We thank the reviewer for pointing this out. We will thoroughly revise the reference section to ensure consistent formatting across all entries, including the cited arXiv papers.
>
> We thank the reviewer again for the positive and constructive feedback. We believe these clarifications will further strengthen the presentation and better align the paper’s terminology and efficiency discussion with its actual scope.

---

> > ### Author Rebuttal · Reviewer_6y98 · 2026-04-02
> >
> > I thank the authors for their effort, and I will maintain my score.

---

> > > ### Author Response · Authors · 2026-04-03
> > >
> > > We are glad that the provided explanations addressed the concerns. Thank you for your careful review and positive feedback.

---

### Official Review · Reviewer_GHud · 2026-03-13

**Soundness:** 3
**Presentation:** 3
**Significance:** 3
**Originality:** 3
**Overall Recommendation:** 4
**Confidence:** 4

**Summary:**

This paper presents BECRA, a training paradigm that evolves forecasting agents by exploring diverse toolsets and distilling the results into interpretable strategic lessons. the authors claim that by verifying these lessons through controlled comparisons and storing them in a dedicated library, the agent can adapt to new time-series tasks in a zero-shot manner without requiring additional training.

**Compliance With Llm Reviewing Policy:**

Affirmed.

**Final Justification:**

The rebuttal is helpful. I reinforce my prior assessment.

**Key Questions For Authors:**

What is the LLM API call and other potential hidden costs? for a typical paper this might not be as important yet since the authors are claiming economic effectiveness (up to 80% reduced costs) it is critical to reveal the `actual' total costs including fees like API calls.

**Limitations:**

Yes

**Strengths And Weaknesses:**

Overall I like this paper especially the fact that it has a real world usage for llm agents with real world cost saving:

- as far as I understand this paper proves away from treating LLMs as raw predictors and instead using them as "strategy planners" for time-series pipelines is a smart, highly practical shift. It directly addresses the brittleness of static forecasting pipelines.

- the traditional AutoML spits out a pipeline with zero explanation. BECRA produces human-readable "causal lessons" (e.g., "This strategy succeeds when seasonal structure is clear and missingness is low..."). This is a massive win for industrial applications where engineers need to trust and debug the system.

- the authors also claim that BECRA reduces trial costs by 80% compared to a cold-start AutoML framework is a compelling result. By amortizing the search cost into a reusable memory bank, it solves a real scalability problem. I also like that the authors didn't just test on clean data but also intentionally injected missing blocks and anomalies and proved their framework degrades much more gracefully than static baselines.

therefore im leaning towards an overall positive rating for this paper with a weak accept. that said I hav a main concern about this paper about the term 'casual' used across the paper that my understanding is the community is notoriously strict about the term "causal." While the authors clarify that their causality is at the "agent decision-making" level (essentially A/B testing an intervention on the prompt), it is fundamentally just robust, A/B-verified conditional heuristics. Calling it "Causal Reasoning" in the title might overstate the theoretical rigor, as it is not performing true causal inference on the underlying data-generating process. as a result the `the causal verification' remains still a bit unclear to me and it is hard to isolate the core contributions regard this.

---

> ### Author Rebuttal · Authors · 2026-03-31
>
> We sincerely thank reviewer GHud for the positive and thoughtful assessment, especially for recognizing BECRA’s practical value in using LLMs as strategy planners, producing human-readable lessons, reducing repeated search cost, and showing stronger robustness under missingness and anomaly corruption. Below we address the two concerns.
> ## Main Concern: The use of "causal" terminology
> We appreciate the reviewer for raising this point and fully agree the strict standards for causal inference, which should be interpreted carefully. As stated in Section 3.3, BECRA does not perform formal causal inference on the data-generating process. The causal interpretation in our paper is instead at the agent decision-making level.
>
> Our verification stage (Algorithm 3) instantiates Woodward's interventionist account of causation [1]. Specifically, each lesson $\phi_k$ is evaluated by conducting controlled interventions on the agent's decision policy: all execution conditions remain unchanged, only the presence or absence of $\phi_k$ is manipulated. This satisfies Woodward's definition of an ideal intervention. A lesson is retained only if $\Delta_{\phi_k} > \delta$ holds across paired rollouts, which corresponds to the invariance requirement that a genuine causal relationship should remain stable under repeated interventions. We additionally verify negative lessons by forcing the agent to contradict them, thereby exploring counterfactual dependence in both directions, which goes beyond standard A/B testing.
>
> We agree, however, that the term "Causal Reasoning" in the title might suggest a stronger claim about casual inference on the data-generating process. To address this, we will clarify this terminology more explicitly in the introduction and method section, emphasizing that our use of “causal” refers to **controlled intervention on agent policy decisions**, rather than causal discovery over the underlying data-generating mechanism.
>
> [1] Woodward, J. (2005). Making things happen: A theory of causal explanation. Oxford university press.
> ## Key Question: LLM API costs and hidden costs
> We appreciate this important question, it reflects exactly the scrutiny that cost-efficiency claims should undergo. We agree that the actual cost structure should be made more explicit.
>
> **Cost of LLM API.**
> - Our tool library (Appendix A.2) covers 6 pipeline stages with 35 tools in total. Based on our pipeline, a complete BECRA deployment (warm-start exploration plus transfer to new datasets) involves LLM calls across 3 stages: lesson extraction (§3.2), lesson verification (§3.3), and lesson-guided planning (§3.4). In the scenario of warm-start exploration plus transfer to new datasets, the LLM API cost is approximately $28–35 under GPT-4 pricing.
> - Moreover, as shown in our transferability analysis (Table 4), the lesson memory of BECRA can be successfully leveraged by open-source LLMs such as LLaMA-3.1-70B and Qwen-2.5-72B with only marginal performance degradation. This indicates that API costs can be completely eliminated by deploying open-source LLMs locally, thereby reducing the external monetary cost to zero.
>
> **Hidden GPU costs of AutoML.**
> - To clarify, the economic effectiveness we claim primarily refer to the reduction in the number of toolchain trials. Each trial is a complete GPU train-evaluation cycle, which is actually a potential economic cost. In industrial environments, organizations maintain hundreds of evolving time series pipelines (e.g., demand forecasting across product lines, energy load prediction across regions). Most of these organizations do not maintain dedicated GPU clusters for pipeline optimization. Instead, they purchase cloud GPU instances or provision additional infrastructure to accelerate training, both of which will generate a large amount of recurring costs. Therefore, each toolchain trial incurs a direct economic cost. AutoML needs to conduct 1177 trials on 6 datasets (Table 5), and each time a new dataset is used, it requires starting from scratch with the full budget. For organizations with 100 datasets, this means approximately 19600 cold-start trials are needed, and each trial consumes GPU time, energy, and engineering maintenance.
> - BECRA changes this cost structure through its reusable lesson memory. After a one-time warm-start of 158 trials, the accumulated lessons can be transferred to all subsequent datasets. Each new task only requires approximately 10 trials. For 100 datasets, BECRA requires approximately 1148 trials, while AutoML requires approximately 19600 trials, reducing by 94%. Our method acquires this transferable knowledge at a low API cost, which in turn avoids thousands of redundant GPU trials.
>
> We will add this cost breakdown and discussion in the revision for greater transparency.
>
> We thank the reviewer again for the encouraging and constructive feedback. We believe these clarifications will make the scope of the causal terminology and the cost-efficiency claim more precise.

---

> > ### Author Rebuttal · Reviewer_GHud · 2026-04-04
> >
> > Thank you for your detailed rebuttal. I retai my current assessment.

---

### Official Review · Reviewer_XGwo · 2026-03-13

**Soundness:** 2
**Presentation:** 3
**Significance:** 3
**Originality:** 3
**Overall Recommendation:** 4
**Confidence:** 3

**Summary:**

This paper proposes BECRA, an agent training framework for adaptive time-series forecasting. Instead of dataset-specific pipeline engineering or employing autoML, this work train a forecasting agent to accumulate reusable symbolic strategy lessons. The framework consists of four stages including contrast-aware UCB exploration, contrastive lesson induction, intervention-based lesson verification, and incorporating learned lessons to downstream datasets through in-context planning. Experimental results suggest that the proposed method achieves competitive forecasting performance while also reducing search cost compared to conventional AutoML approaches.

**Compliance With Llm Reviewing Policy:**

Affirmed.

**Final Justification:**

The authors provided addtional evaluation and analysis which have addressed most of my concerns and convinced me to raise my score.

**Key Questions For Authors:**

See weaknesses.

**Limitations:**

yes

**Strengths And Weaknesses:**

### Strengths
- The main strength of this paper is its novel paradigm. It frames adaptive forecasting as training an agent to acquire reusable symbolic strategy lessons, which can then transfer to unseen datasets. This is a creative and meaningful departure from standard forecasting-model design or per-dataset search.
- The method design is coherent and well motivated. The different stages of the framework connect naturally, and the overall pipeline is conceptually clean.
- The use of symbolic lessons is appealing from the perspective of interpretability and transferability.

### Weaknesses
- The main weakness is that the empirical analysis is not sufficiently thorough relative to the paper’s ambitious claims. As a result, while the proposed paradigm is interesting, the current evidence is still somewhat insufficient to fully validate its reliability and scope.
  - The ablation study is not complete or fine-grained enough, making it hard to clearly isolate the contribution of each component.
  - The paper lacks a careful analysis of agent stability and robustness, such as variance across runs, sensitivity to the underlying LLM, and consistency of learned lessons/planning behavior.

If the authors can address these issues, I would be inclined to raise my score.

---

> ### Author Rebuttal · Authors · 2026-03-31
>
> We sincerely thank reviewer XGwo for recognizing the novelty, coherence, interpretability, and transferability of our framework. We appreciate the constructive suggestion that stronger empirical analysis would further strengthen the paper. Below we address the two weaknesses.
>
> ## Weakness 1: The ablation study does not clearly isolate the contribution of each component
> We agree that a fine-grained ablation study is important. To achieve this, we implemented the isolation of each component, which includes **seven ablation conditions spanning four distinct axes**, corresponding to the four stages of BECRA. In addition, Appendix A.5 (Figure 3) provides a complementary mechanism analysis by varying **meta-feature availability** and **tool diversity** separately.
>
> Here we reorganized Table 3 for the ablation settings.
>
> #### Overview of ablation components (reorganized from Table 3 in our paper)
> | Axis | Variant | What is changed |
> |---|---|---:|
> | Full model | **BECRA** | — |
> | Symbolic lessons | w/o Lessons | Remove lesson-guided planning entirely |
> | Exploration | Greedy | Replace contrast-aware UCB with greedy sampling |
> | Verification | w/o Verification | Skip the lesson verification stage |
> | Knowledge use | LLaMA-7B | Replace ICL with SFT |
> |  | ChatGLM-6B | Replace ICL with SFT |
> |  | Mistral-7B | Replace ICL with SFT |
>
> These ablations lead to following component-wise findings from Table 3:
> - Symbolic lessons are the primary source of transferable planning ability, as removing lessons causes the largest performance drop.
> - Contrast-aware exploration and verification each contribute to reliability, since both replacing UCB with greedy exploration and removing verification degrade performance.
> - ICL is more effective than SFT under limited supervision, as replacing ICL with SFT consistently reduces performance across all tested backbones.
>
> Appendix A.5 further clarifies the mechanism behind these results. When meta-features are progressively removed, the agent can still generate candidate lessons, but the number of verified lessons drops sharply. This suggests that the meta-features are mainly important for making lessons sufficiently specific and verifiable. In contrast, when toolchain diversity is reduced, both extracted and verified lessons decrease, indicating that diverse toolchains provide the behavioral contrast needed for both lesson induction and validation.
>
> **In the revision, we will discuss these component-wise findings more explicit and incorporate the key observations from Appendix A.5 into the main abalation discussion.**
>
> ## Weakness 2: Lack of agent stability and robustness analysis
> We agree that stability and robustness should be presented more explicitly. Our paper contains supporting evidence in this regard, which we summarize below from three perspectives: variance across runs, sensitivity to the underlying LLM, and consistency of planning behavior.
>
> ### 2.1 Variance across runs
> Appendix A.7 (Table 6) reports **95% confidence intervals over 5 independent runs (using different random seeds)**. BECRA's MOE remains comparable to standard trainable baselines such as TimeMixer and TimesNet, and is even smaller on some datasets (e.g. ETTm1 and Electricity). This suggests that the multi-stage agent pipeline does not introduce abnormal inter-run instability.
>
> ### 2.2 Sensitivity to the underlying LLM
> Table 4 presents a direct sensitivity test in which the **lesson memory and prompt structure are fixed**, while only the reasoning engine (LLM) is changed. Compared to the default GPT-4 setting, the average MSE changes remain very small: **Claude 3.5 (+0.6%)**, **Qwen-2.5-72B (+1.3%)**, **Gemini Pro (+1.6%)**, and **LLaMA-3.1-70B (+2.9%)**. This indicates that the learned symbolic lessons remain effective across different strong LLMs, and that BECRA is not  coupled to a single reasoning engine.
>
> ### 2.3 Consistency of planning behavior
> To further assess behavioral consistency, we conducted an additional analysis on 20 verified lessons (see table below). **For each lesson**, we ran Lesson-Guided Planning process on all datasets whose meta-features satisfy the lesson's conditions. This yields 33 (lesson, dataset) pairs in total. **For each pair**, we record whether the agent's toolchain selection conforms to the lesson's recommendation. The 90.9% overall adherence rate indicates that verified lessons guide downstream planning in a highly consistent manner.
>
> | Lesson Type | Lessons | (Lesson, Dataset) Pairs | Adherent | Adherence Rate |
> |---|---|---|---|---|
> | Positive | 12 | 21 | 19 | 90.5% |
> | Negative | 8 | 12 | 11 | 91.7% |
> | **Overall** | **20** | **33** | **30** | **90.9%** |
>
> We will revise the paper to present these stability results more directly, and explicitly connect the evidence from Table 4, Table 6, and consistency analysis in the main discussion.
>
> We hope these clarifications address the reviewer’s concerns regarding empirical completeness, component isolation, and robustness.

---

> > ### Author Rebuttal · Reviewer_XGwo · 2026-04-03
> >
> > Thank you for the detailed rebuttal. I appreciate the additional empirical evidence which has addressed most of my concerns. I have raised my score to 4.

---

> > > ### Author Response · Authors · 2026-04-03
> > >
> > > It is encouraging to know that our supplementary analyses successfully resolved your concerns. We sincerely appreciate your constructive advice and support.

---

### Decision · Program_Chairs · 2026-04-30

**Decision:**

Accept (regular)

**Comment:**

This paper proposes BECRA, a training paradigm for adaptive forecasting agents that learns reusable symbolic strategy lessons through exploration, verification, and lesson-guided planning, with the goal of enabling zero-shot adaptation across time-series datasets. Reviewers generally agreed that this is an interesting and practically meaningful direction, and they found the overall framework coherent, interpretable, and well motivated. The empirical results were also viewed positively, especially the evidence that the learned lessons can improve adaptation performance while reducing repeated search cost.

The main concerns centered on whether the empirical analysis was sufficiently complete for the paper’s broader claims, including finer-grained ablations, robustness and stability analysis, cost accounting, and the use of the term “causal reasoning.” Some reviewers also noted remaining limitations such as reliance on handcrafted meta-features and nontrivial upfront exploration cost.

The rebuttal largely addressed many of the concerns by clarifying the scope of the causal claim, adding more explicit ablation, stability, consistency, and efficiency analyses, and providing additional discussion of failure cases and lesson behavior. Overall, while the method still has some practical limitations, the paper presents a clearly scoped and well-supported contribution that reviewers overall judged to be useful and promising. On balance, I recommend acceptance.